# The Centennial Collection of VDR Ligands: Metabolites, Analogs, Hybrids and Non-Secosteroidal Ligands

**DOI:** 10.3390/nu14224927

**Published:** 2022-11-21

**Authors:** Miguel A. Maestro, Samuel Seoane

**Affiliations:** 1Department of Chemistry-CICA, University of A Coruña, Campus da Zapateira, s/n, 15008 A Coruña, Spain; 2Department of Physiology-CIMUS, University of Santiago, Campus Vida, 15005 Santiago, Spain

**Keywords:** metabolites, analogs, hybrids and VDR nonsecosteroidal ligands

## Abstract

Since the discovery of vitamin D a century ago, a great number of metabolites, analogs, hybrids and nonsteroidal VDR ligands have been developed. An enormous effort has been made to synthesize compounds which present beneficial properties while attaining lower calcium serum levels than calcitriol. This structural review covers VDR ligands published to date.

## 1. Introduction

Since the chemical structure of vitamin D_3_ (**1**, Figure 1 [1,2,3,4,5,6,7,8,9,10,11,12,13,14,15,16,17,18,19,20,21,22,23,24,25,26,27,28,29,30,31,32,33,34,35,36], cholecalciferol) was established in 1932, successive studies have shown it to be essential in physiological processes. Two hydroxylations of **1** are necessary before attaining its most biologically active form. The first is a 25-hydroxylation, which occurs mainly in the liver and produces the most abundant circulating metabolite, 25-hydroxyvitamin D_3_ (**11**, Figure 1, 25-hydroxycholecalciferol, calcidiol, 25OHD_3_) [12]. Subsequently, a second hydroxylation at the 1α position generates the vitamin D hormone, 1α,25-dihydroxyvitamin D_3_ (**13**, Figure 1, 1α,25-dihydroxycholecalciferol, calcitriol, 1,25(OH)_2_D_3_) [14]. This is a pleiotropic hormone that exerts genomic actions by binding to its specific receptor (the vitamin D receptor, VDR), which is present on target cells and found in more than 200 different tissues.

The biological role of 1,25(OH)_2_D_3_ has been related to calcium and phosphorus homeostasis. However, the effects of vitamin D are not limited to mineral homeostasis, skeletal health maintenance, or immune modulation. In addition, this hormone also has fundamental effects on cellular proliferation and differentiation, regulating genes involved in the cell cycle and apoptosis both in normal and tumor cells. These properties and its wide distribution have led to the study of its effects on various pathologies, such as osteoporosis and cancer, thus arousing interest in the field of health and the pharmaceutical industry. Unfortunately, the therapeutic use of 1,25(OH)_2_D_3_ also leads to an increase in the concentration of calcium in blood (hypercalcemia), which can cause significant side effects. Therefore, numerous attempts have been made to synthesize noncalcemic analogs of 1,25(OH)_2_D_3_ for use in health treatment.

In recent decades, structure–function relationships (SARs) have been determined to support the chemical modifications of the secosteroid structure of 1,25(OH)_2_D_3_. The novel structures’ goal is to reduce their calcemic activity in comparison with calcitriol while exerting their interesting biological properties. A huge synthesis effort has been carried out, yielding interesting chemical reviews in this regard [2]. The current review updates the scientific information on the structural library of VDR ligands and incorporates nonsteroidal VDR ligands.

## 2. Materials and Methods

All compounds contained in this review were collected from published papers and patents. Most of the materials were freely accessible via the Internet, and paper copies were available in other cases. After careful reading, relevant structures were drawn using CHEMDRAW software [3]. No database was generated. A structural analysis of this collection may require future elaboration of a database.

## 3. Results

We found 1778 VDR compounds, which are displayed chronologically in 31 figures. All of these compounds are ligands that specifically bind to their VDR receptor. This binding allows the interaction of the 1,25(OH)_2_D_3_-VDR complex with target genes in the cell nucleus, modulating their expression and mediating a biological response. The following color scheme was used in the figures: dark blue corresponds to marketed compounds (Figures 1, 3, 4, 8 and 9), light violet to outstanding compounds with interesting properties (Figures 2, 4–10, 12–15, 17–21 and 23–31), and dark green to non-secosteroidal VDR ligands (Figures 9, 11, 12, 15 and 20–27).

Vitamin D is closely associated with calcium and phosphorus homeostasis. No scientific rational has yet been found for the calcemic properties of a compound in comparison with calcitriol. Therefore, structure–function relationships (SARs) were carried out in order to validate the key modifications in the structure of 1,25(OH)_2_D_3_ that may alter biological and calcemic properties. After more than 50 years of study, some hints have been obtained. For example, it is known that C-19 methylene deletion yields low calcemic analogs; it is also known that deletion/substitution of the steroidal cycles de-A ring, de-C ring, and/or de-D ring may yield low calcemic analogs. Lowering the calcemic side effects of the vitamin D analogs is important; however, we must not lose sight of other modifications that may increase the antiproliferative and prodifferentiation activity (side-chain modification with extra double and or triple bonds) as well as increase the metabolic stability (fluorine atom incorporation). In summary, the following main structural topics are covered in the current review:C-21 Methyl epimerization;C-19 Methylene deletion;Incorporation of fluorine atoms;Deletion/substitution of steroidal cycles: de-A ring, de-C ring, and/or de-D ring;C-2 Functionalization;C-3 Epimerization;Side-chain modification with extra double and/or triple bonds, heteroatoms, and/or branched hydrocarbons.

What is novel in this collection is the incorporation of non-secosteroidal VDR ligands (dark green). In 1999, Boehm [4] hypothesized that “non-secosteroidal VDR ligands might display different profiles of activity and metabolism than do secosteroidal l,25(OH)_2_D_3_, analogs, including less calcemic properties, which might render them attractive as both topical and oral pharmaceuticals for treating a variety of diseases. This hypothesis was based in part on the success that nonsteroidal androgen receptor (AR) and estrogen receptor (ER) modulators have had as drugs. Nonsteroidal compounds have been synthesized that modulate the activity of these receptors and show enhanced tissue selectivity in comparison to the steroids”.

Figure 1 (1931–1978) [1,2,3,4,5,6,7,8,9,10,11,12,13,14,15,16,17,18,19,20,21,22,23,24,25,26,27,28,29,30,31,32,33,34,35,36]. Vitamin D_3_ (**1**, cholecalciferol) [1] was discovered in 1922, but it was not chemically characterized until 1931. Dihydrotachysterol_2_ (**5**) [10] was introduced in 1934, and it is still on the market as an antitetanic agent AT-10. In 1968, the most abundant metabolite of vitamin D_3_ was discovered as 25-hydroxyvitamin D_3_ (**11**, 25-hydroxycholecalciferol) [18], and in 1971, 1α,25-dihydroxyvitamin D_3_ **13**, 1α,25-dihydroxycholecalciferol, calcitriol, 1α,25(OH)_2_D_3_ [21], the vitamin D_3_ hormone, was identified. Later, 1α-hydroxyvitamin D_3_ (**21**, Alfacalcidiol) [25], a synthetic analog, was marketed for the treatment of secondary hyperparathyroidism (2HPT), renal failure, and osteoporosis.

Figure 2 (1978–1982) [37,38,39,40,41,42,43,44,45,46,47,48,49,50,51,52]. 25-Hydroxyvitamin D_3_ 26(23)-lactones (**58**–**61**) were discovered in 1980 [50,51,52], and they behave as antagonists of gene transcription induced by VDR. They were the first compounds discovered to have antagonist properties.

Figure 3 (1982–1987) [53,54,55,56,57,58,59,60,61,62,63,64,65,66,67,68,69,70,71,72,73,74,75,76,77,78]. 26,26,26,27,27,27-Hexafluoro-1α,25-dihydroxyvitamin D_3_ (**70**, Falecalcitriol) [60] is used in the treatment of 2HPT and osteoporosis. 1α,25-Dihydroxy-22-oxavitamin D_3_ (**100**, Maxacalcitol) [76] is used in 2HPT and psoriasis.

Figure 4 (1987–1991) [78,79,80,81,82,83,84,85,86,87,88,89,90,91,92,93,94]. **111** (Calcipotriol, MC903) [79] is marketed as a treatment with exceptional clinical response in psoriasis. 1α,25-Dihydroxy-22(23)-didehydrovitamin D_3_ (**116**) [83] has shown potent antiproliferative activity. 2β-(Hydroxypropoxy)-1α,25-dihydroxyvitamin D_3_ (**131,** ED-71) [85] is used in osteoporosis treatment.

Figure 5 (1991–1992) [95,96,97,98,99,100,101,102,103,104,105,106,107,108,109,110,111,112,113,114,115,116,117]. Compound **186** [107] is an important analog functionalized at C-11 that may allow the synthesis of haptens, without disturbing the VDR ligand anchoring groups (1α-OH, 3β-OH and 25-OH).

Figure 6 (1993–1994) [118,119,120,121,122,123,124,125,126,127,128,129,130,131,132,133,134,135,136]. Compounds **225** [93] and **208** [108] were independently developed by different research groups and are important analogs functionalized at C-18 and C-11, respectively. They may allow the synthesis of haptens without disturbing the VDR ligand anchoring groups.

Figure 7 (1994–1997) [136,137,138,139,140,141,142,143,144,145,146,147,148]. Compounds **308** and **309** [147] present an interesting property by exhibiting only nongenomic rapid effects at physiological concentrations. Moreover, 1α-hydroxyl group addition (**309**) does not alter the sensitivity of nongenomic effects of **308**.

Figure 8 (1997–1999) [149,150,151,152,153,154,155,156,157,158]. 1α-Hydroxyvitamin D_2_ (**325**, Doxercalciferol) [151] is marketed as a 2HPT treatment. (22*E*,24*E*)-Diene-24,26,27-trishomo-19-nor-1α,25-dihydroxyvitamin D_3_ (**348**, Ro 25-8584) [152] represents an outstanding compound inhibiting the proliferation in myeloid leukemia cell lines. When 2-methylene-19-nor-1α,25-dihydroxyvitamin D_3_ (**349**, 2MD) [156] is given as oral therapy, it is at least 100 times more potent than 1α,25(OH)_2_D_3_ in stimulating bone mass increase. A randomized clinical trial showed that **349** increased bone turnover but not BMD (bone mass density) in postmenopausal woman with osteopenia.

Figure 9 (1999) [158,159,160,161,162,163,164,165,166,167,168]. 24*R*,25-Dihydroxyvitamin D_3_ (**388**, Tacalcitol) [160] is prescribed for psoriasis. 24,26,27-Trishomo-1α,25-dihydroxyvitamin D_3_ (**406**, Seocalcitol, EB 1089) [163] acts as a powerful antiproliferative used in breast, colon, or pancreas tumor models.

Figure 10 (2000–2001) [169,170,171,172,173,174,175,176,177,178,179,180,181,182]. 1α-Hydroxy-26(27)-dehydro-25-(butylcarboxylate)-vitamin D_3_ (**433**, ZK159222) and 1α-hydroxy-26(27)-dehydro-25-(ethylpropenoate)-vitamin D_3_ (**434**, ZK168281) [170] have been identified as VDR antagonists, though **434** is more potent than **433**. Both compounds selectively stabilize an antagonist conformation of the VDR-LBD (ligand-binding domain). 1α,25-Dihydroxy-21-(3-hydroxy-3-methylbutyl)-vitamin D_3_ (**435**, Gemini) [171] has emerged as the lead compound with superior gene transcription activity and tumor-cell-line inhibition.

Figure 11 (2001–2002) [183,184,185,186,187,188,189,190,191,192,193,194,195,196]. 1α,25-(OH)_2_-16-ene-20-epi-23-yne-3-epi-D_3_ (**493**), 1α,25(OH)_2_-16-ene-23-yne-hexafluoro-3-epi-D_3_ (**494**), and 1α,25(OH)_2_-16-ene-3-epi-D_3_ (**495**) are potent inducers of apoptosis of HL-60 cells. Their 3-natural (3β-OH) analogs have been shown to be potent modulators of HL-60 cell growth and differentiation [184]. This is the first report to demonstrate that the epimerization of the hydroxyl group at C-3 of the A-ring of 1α,25(OH)_2_D_3_ plays an important modulation role for HL-60 cell differentiation and apoptosis. 2,2-Difluoro-1α,25-dihydroxyvitamin D_3_ (**507**) [185] is similar to 1,25(OH)_2_D_3_ in terms of in vitro antiproliferative activity, but it is different in terms of transcriptional activity. In addition, **507** is 2–3 times more transcriptionally active than calcitriol, with similar in vivo calcemic activity. 2,2-Dimethyl-1α,25-dihydroxy-19-norvitamin D_3_ (**509**) [186] is 7.5 times less transcriptionally active than calcitriol and considerably less calcemic. Moreover, **509** strongly suppresses parathyroid hormone (PTH) secretion.

Figure 12 (2002) [197,198,199,200,201,202,203,204]. Seco-C-9,11-bisnor-17-methyl-26,26,26,27,27,27-hexafluoro-20-epi-1α,25-dihydroxyvitamin D_3_ (**533**, WY1112) [197] and seco-C-9,11,21-trisnor-17-methyl-23(24)-didehydro-26,26,26,27,27,27-hexafluoro-1α,25-dihydroxyvitamin D_3_ (**559**, CD578) [198] display high differentiation ratios between antiproliferative and calcemic affects. 26,27-Bishomo-1α-fluoro,25-hydroxy-23-en-vitamin D_3_ (**582**, Ro-26-9228) [203] is used for treatment of osteoporosis.

Figure 13 (2003–2004) [205,206,207,208,209,210,211,212,213,214,215,216,217,218]. Dienyne **646** [215] represents the first locked side-chain analog of calcitriol with remarkable VDR transcriptional activity. Lactone **657** [217] showed one order of magnitude higher antagonist activity than lactone **66** (Figure 2).

Figure 14 (2004–2006) [218,219,220,221,222,223]. Further development in double side-chain vitamin D analogs, the Gemini series, made it possible to assess the steric VDR requirements of drug candidates. Compounds **684–695 [220]** present two different side chains at C-20 that improve their toxicity profiles and pharmacokinetic drug performance.

Figure 15 (2006–2007) [224,225,226,227,228,229,230,231,232,233,234,235,236,237,238,239,240]. C-20 cyclopropyl vitamin D_3_ analog **755** [233] showed high MLR (mixed lymphocyte reaction) activity for the suppression of interferon-γ release with no calcemic activity. Immunomodulatory activity was measured by suppression of interferon-γ release in mixed lymphocyte reaction cells. The inhibition of clonal proliferation was evaluated in the leukemia HL-60, breast cancer MCF-7, prostate PC-3, and LNCaP cell lines. Significant separation of the immunomodulatory activity from hypercalcemic effects (MTD, maximum tolerated dose) was observed. Compound **747** was 2900 times more active and 100 times less hypercalcemic than 1α,25(OH)_2_D_3_, while **755** was 29 times more active and 100 less hypercalcemic. In the breast cancer MCF-7 cell line, compounds **753**, **754**, **755**, and **757** were ten thousand times more active but equally or less hypercalcemic than 1α,25(OH)_2_D_3_. Metabolism of 16-ene-20-cyclopropyl compounds is arrested at the 24-keto stage, which explains the increased biological activity of the 16-ene variants.

Figure 16 (2006–2008) [241,242,243,244,245,246,247,248,249,250,251,252,253]. Intensive research activity was carried out on the leading structures with outstanding biological properties, i.e., Gemini compounds **799–803** [246,247]. These studies focused on the structural modifications of Gemini that influenced the differentiation-inducing, antiproliferative, and transcriptional activity of the compounds in human leukemia cells. The cyclopropyl modification at the pro-*R* side chain decreased the activity of the compound compared to 1α,25(OH)_2_D_3_, and further A-ring modifications did not restore this activity. Cyclopropyl modification at the pro-*S* side chain of Gemini increased the VDR-induced transcriptional activity. In addition, privileged VDR antagonists lactones **804–832** and **833**–**864** [243,244] were studied. The antagonistic activity was markedly affected by the structure of the lactone ring, including length of the alkyl chain and the stereochemistries on the C23 and C24 positions. The VDR binding affinity of the (23*S*,24*S*)-24-alkylated vitamin D_3_ lactones increased 2.3–3.7-fold as compared to the unsubstituted lactones **64**–**67** (Figure 2). The antagonistic activity of (23*S*,24*S*)-isomers were enhanced to be 2.2-,3.5-, 1.8-, and 1.7-fold higher compared to the unsubstituted lactones **64**–**67** (Figure 2).

Figure 17 (2008–2009) [254,255,256,257,258,259,260,261,262,263,264]. 2-Methylene-19-nor-(20*S*)-1α-hydroxy-bishomopregnacalciferol **942** [20(*S*)-2MbisP] [263] were able to suppress PTH at levels that did not stimulate bone resorption, intestinal calcium, or phosphate absorption and may have potential for use in the treatment of 2HPT in chronic kidney disease.

Figure 18 (2009–2010) [265,266,267,268,269,270]. Hybrid compounds **1020** (26,27-bis-nor-25-bishomo-19-nor-25’-oxo-25”-methylcarboxamide-1α-hydroxyvitamin D_3_) and **1022** (26,27-bis-nor-25-homo-19-nor-25’-(2aminophenyl)-carboxamide-1α-hydroxyvitamin D_3_) [270] showed antiproliferative activity against AT84 carcinoma cells; neither of them induced hypercalcemia even at concentrations 100-fold higher than those tolerated for 1,25D. This demonstrates that it is possible to create a wide range of bifunctional molecules that possess VDR agonism and HDACi (histone deacetylases inhibitor) activity. Structural latitude is significant with a wide variety of ZBGs (zinc-binding group) amenable to incorporation into the side chain of vitamin D-like secosteroids. Importantly, several of these molecules function as antiproliferative agents against AT84 cells in vitro, while possessing minimal hypercalcemic activity in vivo.

Figure 19 (2009–2010) [271,272,273,274,275,276,277,278,279,280,281,282,283]. Intensive research activity was carried out on Gemini compounds **1053**–**1069** [273]. Calcitriol was implicated in many cellular functions including cell growth and differentiation. It was shown that Gemini compounds were active in gene transcription induction with enhanced antitumor activity. Fine tuning of their structurally derived biological properties would be required for therapeutic use.

Figure 20 (2010–2012) [284,285,286,287,288,289,290,291,292,293,294,295,296,297,298]. 25-Diethylphosphite-1α-hydroxy-23(24)-didehydrovitamin D_3_ **1131** [290] was tested for antiproliferative effects on several human and murine tumor cell lines: human squamous cell carcinoma HN12, human glioma T98G, and Kaposi sarcoma SVEC vGPCR cell lines. Furthermore, in human glioma T98G and human squamous cell carcinoma HN12 cell lines, the antiproliferative effects exerted by compound **1131** were greater than those elicited by 1α,25(OH)_2_D_3_. Visual observation of internal animal organs such as liver, duodenum, lungs, and kidneys showed no macroscopic morphological alterations after treatment with this compound. This compound appears to be well tolerated even at high doses. Altogether, these results suggest that compound **1131** exerts considerable antiproliferative activity at nonhypercalcemic dosages and may have therapeutic potential for the treatment of various hyperproliferative disorders. Non-secosteroidal VDR ligand (4-{1-ethyl-1-[4-(2-hydroxy-3,3-dimethyl-butoxy)-3-methyl-phenyl]-propyl}-2-methyl-phenoxy)-hydroxyacetamide **1173** [295] was confirmed to significantly prevent bone loss after daily treatment without inducing hypercalcemia. These types of compound are potent inhibitors of the Hh (Hedgehog) signaling pathway. Studies show that, contrary to secosteroidal hybrids, the optimal location for incorporating the highly hydrophilic hydroxamic acid corresponds to the portion of the molecules that serve as secosteroidal A-ring mimetics. The best hybrid, **1173**, is a full VDR agonist, as assessed by several criteria, and an efficacious antiproliferative agent against both 1,25D-sensitive (SCC25, AT84) and 1α,25(OH)_2_D_3_-resistant (SCC4) squamous carcinoma cell lines. Importantly, the activity in 1α,25(OH)_2_D_3_-resistant SCC4 cells required both the VDR agonism and HDACi activity of **1173**. This study revealed the remarkable flexibility in the conversion of calcitriol analogs into fully integrated bifunctional molecules, suggesting that it may be possible to extend fully integrated bifunctionalization to other pharmacophores. 

Figure 21 (2012–2013) [298,299,300,301,302,303,304,305,306,307,308,309,310,311,312,313]. 24*S*-Methyl-21-epi-2-methylene-22-oxa-1α,25-dihydroxyvitamin D_3_ (**1191**, VS-105) [306] bound to VDR is highly inductive of functional responses in vitro and effectively suppresses PTH in a dose range that does not affect serum calcium in 5/6 NX uremic rats. [6-(4-{1-Ethyl-1-[4-((*E*)-3-ethyl-3-hydroxy-1-pentenyl)-3-methylphenyl]propyl}-2-methylphenyl)pyridin-3-yl]acetic acid (**1218**) [308] showed excellent ability to prevent BMD loss in mature rats in an osteoporosis model, without severe hypercalcemia and with good PK profiling.

Figure 22 (2013–2014) [313,314,315,316]. Compounds 1**247**–**1301** (non-secosteroidal VDR ligands) [315] were analyzed and presented better therapeutic efficacy when compared to 1α,25(OH)_2_D_3_ in experimental models of cancer and osteoporosis with less induction of hypercalcemia, a major potential adverse effect in the clinical application of VDR ligands. Compounds **1302**–**1313** [316] were analyzed for their binding affinity and inhibitory activity against CYP24A1 (24-hydroxylase; this mitochondrial protein initiates the degradation of 1α,25(OH)_2_D_3_ by hydroxylation of the side chain), and the imidazole styrylbenzamides **1305**–**1309** were identified as potent inhibitors of CYP24A1, with similar or greater CYP27B1 (1α-hydroxylase; the protein encoded by this gene it hydroxylates 25OHD_3_ at the 1α-position, producing 1α,25(OH)_2_D_3_) selectivity than standard ketoconazole. Further evaluation of the 3,5-dimethoxy (**1308**) and 3,4,5-trimethoxy derivatives (**1309**) in chronic lymphocytic leukemia cells revealed that cotreatment of 1α,25-dihydroxyvitamin D_3_ and inhibitor upregulated GADD45α (growth arrest and DNA damage 45 gen) and CDKN1A (cyclin-dependent kinase inhibitor 1A gen).

Figure 23 (2014) [317,318,319,320,321]. Intensive research activity was carried out on Gemini compounds **1338**–**1364** [320].

Figure 24 (2014–2015) [322,323,324,325,326,327,328,329,330,331,332,333,334,335,336]. 1α,20*S*,24*R*-Trihydroxyvitamin D_3_ (**1410**) [332] showed a higher degree of activation, anti-inflammatory activity, and antiproliferative activity than vitamin D_3_ receptor.

Figure 25 (2015–2017) [337,338,339,340,341,342,343,344,345,346,347,348,349,350,351]. 1α,25-Dihydroxy-21-(3-hydroxy-3-methyl-1-methylene-butyl)vitamin D_3_ (**1428**, UV1) [337] presented potent antitumoral effects over a wide panel of tumor cell lines without inducing hypercalcemia or toxicity in vivo. The first vitamin D analog carrying an *o*-carborane in the side chain **1436** [340] showed that the substitution of hydroxyl group at C-25 by this apolar bulky group was possible. VDR binding was half of calcitriol’s, the transcriptional activity was similar, and the calcemic induction was significantly lower. **1436** is an outstanding B-carrier containing 10 boron atoms, which notably bind to VDR, a nuclear receptor. This suggests that **1436** may be interesting as a BNCT (boron neutron capture therapy) drug.

Figure 26 (2017–2018) [351,352,353,354,355,356,357,358]. 1,1’-([4-(3-[4-(3-Hydroxypropoxy)-3-methylphenyl]pentan-3-yl)-1,2-phenylene]bis(oxy))bis(3,3-dimethylbutan-2-ol) (**1503**) [358] displayed efficient inhibitory activity against collagen deposition and fibrotic gene expression in chronic pancreatitis. It also showed physicochemical and pharmacokinetic properties with antitumor activity, highlighting its potential therapeutic applications in cancer treatment.

Figure 27 (2018) [359,360,361,362,363,364]. (1*R*,3*S*,*Z*)-5-{(*E*)-3-[3-(6-Hydroxy-6-methylheptyl)phenyl]pent-2-en-1-ylidene}-4-methylenecyclohexane-1,3-diol (**1573**) [359] exhibited significant tumor growth inhibition and increased survival in SCID mouse models implanted with MDA-MB-231 breast tumor cells. Des-C-ring aromatic D-ring analog **1587** [363] showed remarkable lack of calcemic activity together with its significant antiproliferative and transcriptional activities in breast cancer cell lines, suggesting the therapeutical potential of **1587** for the treatment of breast tumors.

Figure 28 (2018–2019) [365,366,367,368,369,370,371,372,373,374,375,376,377,378]. 21-nor-17(*S*)-Methyl-20(22),23(24)-didehydro-26,26,26,27,27,27-hexafluoro-1α,25-dihydroxyvitamin D_3_ (**1600**) [368] bound strongly to VDR ligand binding domain and induced VDR transcriptional activity. Hybrid **1619** [371] was found to be a potent inhibitor of Hh (Hedgehog) signaling pathway.

Figure 29 (2019–2020) [379,380,381,382,383]. It is known that 25(OH)D3, down-regulates SREBP (sterol regulatory element-binding protein) independently of VDR. A screening of over 250 vitamin D congeners was carried out for their ability to inhibit the activity of an SREBP-responsive luciferase reporter. This is a VDR-responsive reporter assay. A comparison of the relative activity of the six compounds revealed **1639 [379]** as the VDR-selective activator.

Figure 30 (2020–2022) [384,385,386,387,388,389]. Des-C-ring aromatic D-ring analogs **1712** and **1713** [373] showed a remarkable lack of calcemic activity together with significant antiproliferative and transcriptional properties in breast cancer cell lines, suggesting a therapeutical potential for **1712** and **1713** in breast tumor treatment.

Figure 31 (2021–2022) [390,391,392]. KK-052 (**1746**) [391], was found to be the first vitamin D-based SREBP (sterol regulatory element-binding proteins) inhibitor that mitigates hepatic lipid accumulation without calcemic action in mice. KK-052 maintained the ability of 25-hydroxyvitamin D_3_ to induce the degradation of SREBP but lacked VDR-mediated activity. KK-052 serves as a valuable compound for interrogating SREBP/SCAP in vivo and may represent an unprecedented translational opportunity for synthetic vitamin D analogs.

**Figure 1 nutrients-14-04927-f001:**
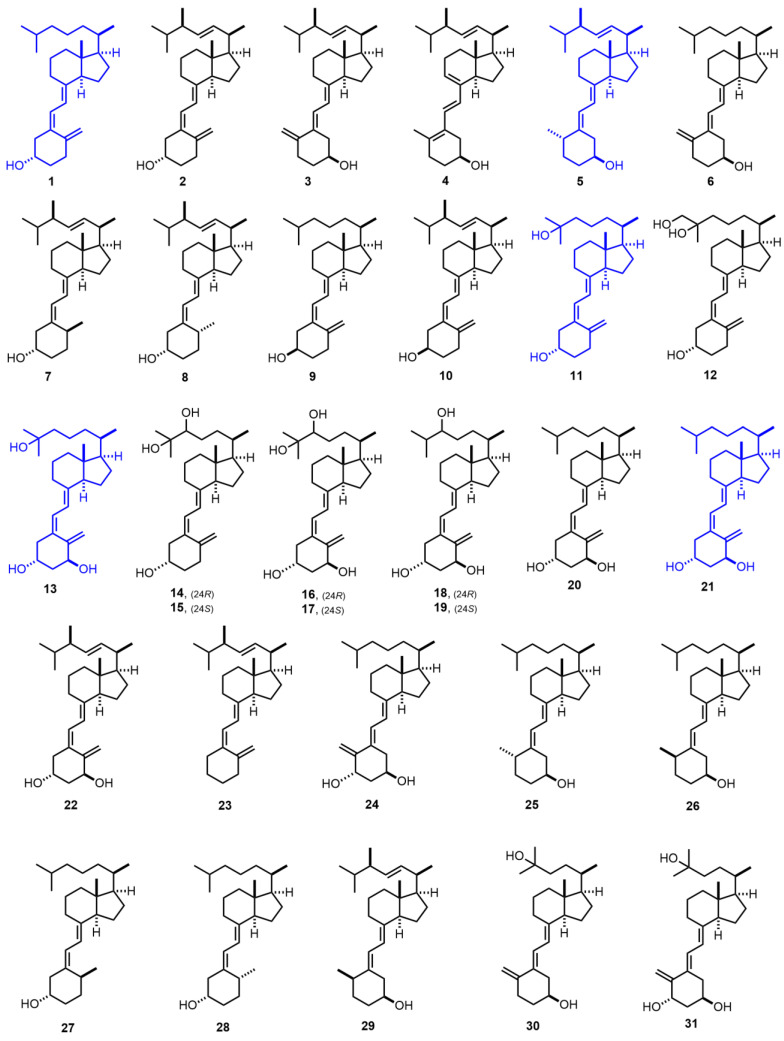
(1931–1978) [1,2,3,4,5,6,7,8,9,10,11,12,13,14,15,16,17,18,19,20,21,22,23,24,25,26,27,28,29,30,31,32,33,34,35,36].

**Figure 2 nutrients-14-04927-f002:**
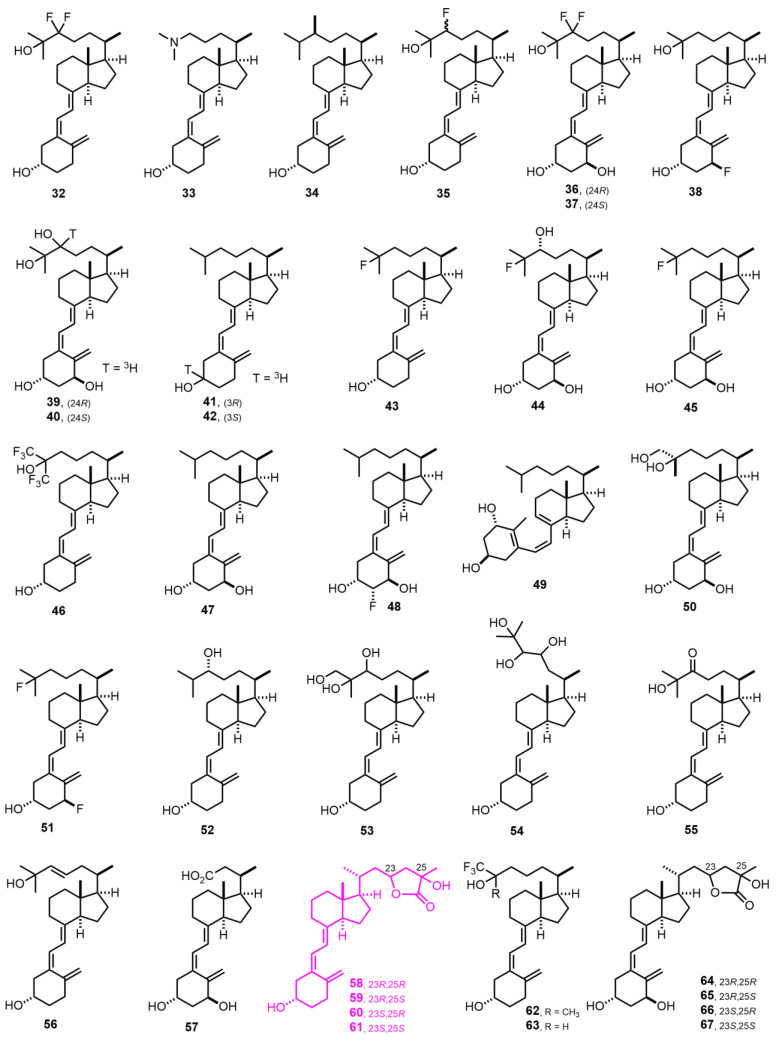
(1978–1982) [37,38,39,40,41,42,43,44,45,46,47,48,49,50,51,52].

**Figure 3 nutrients-14-04927-f003:**
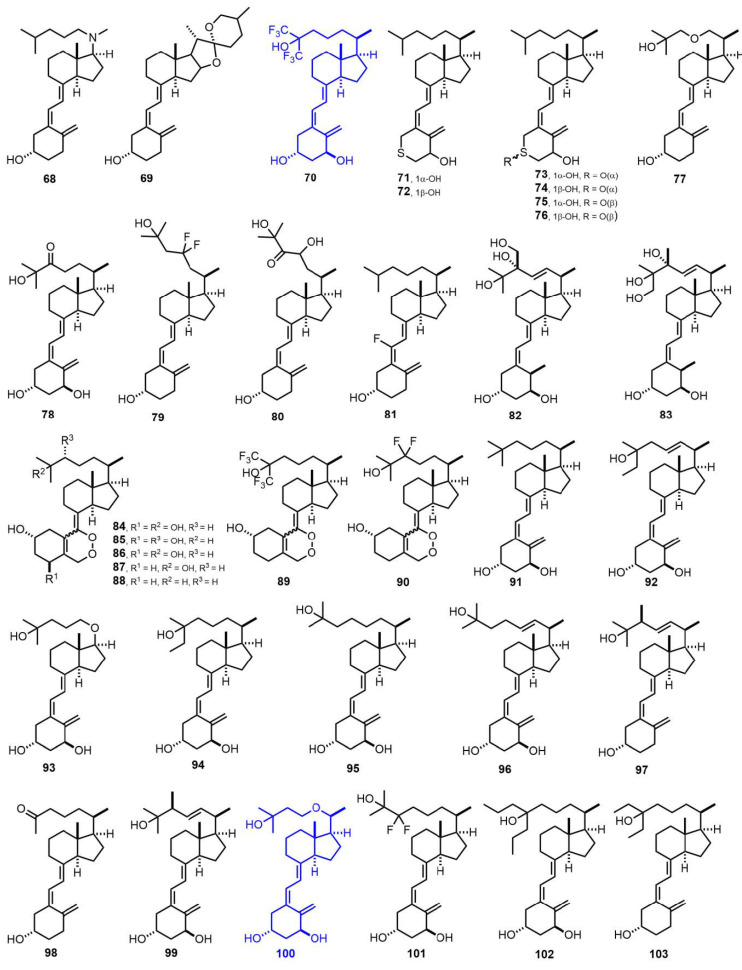
(1982–1987) [53,54,55,56,57,58,59,60,61,62,63,64,65,66,67,68,69,70,71,72,73,74,75,76,77,78].

**Figure 4 nutrients-14-04927-f004:**
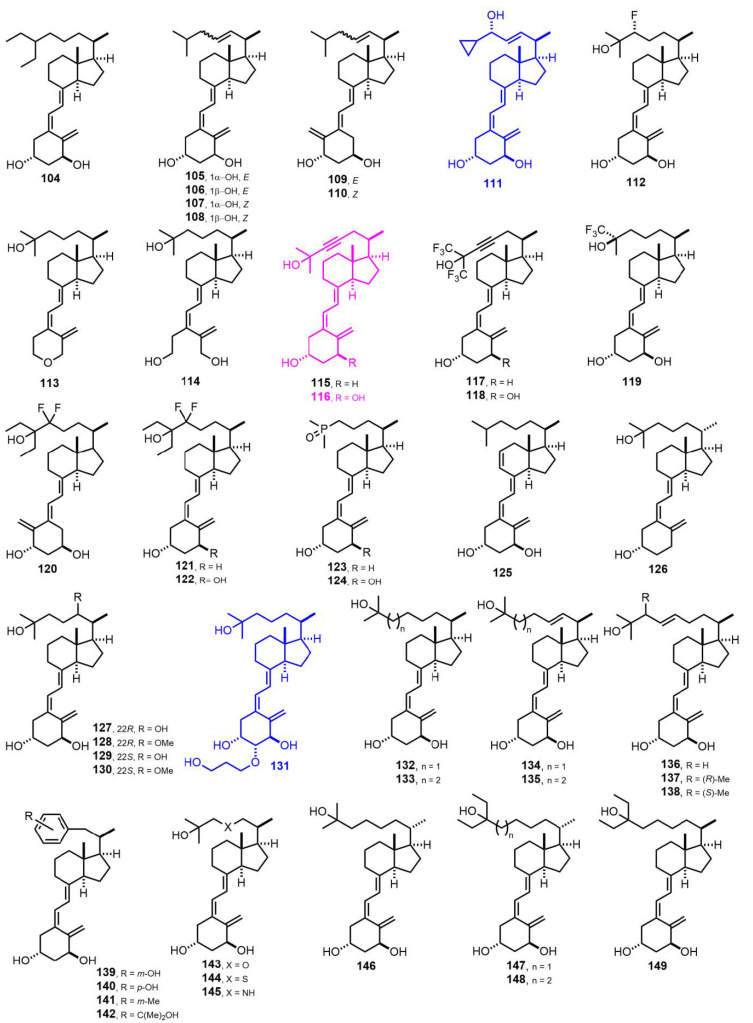
(1987–1991) [78,79,80,81,82,83,84,85,86,87,88,89,90,91,92,93,94].

**Figure 5 nutrients-14-04927-f005:**
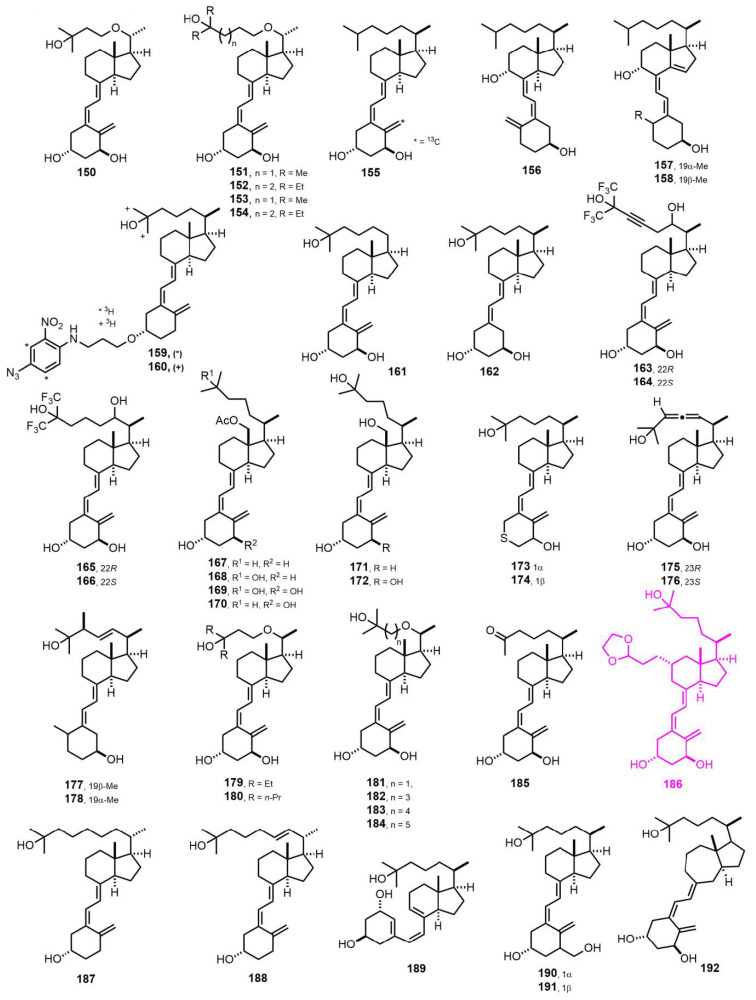
(1991–1992) [95,96,97,98,99,100,101,102,103,104,105,106,107,108,109,110,111,112,113,114,115,116,117].

**Figure 6 nutrients-14-04927-f006:**
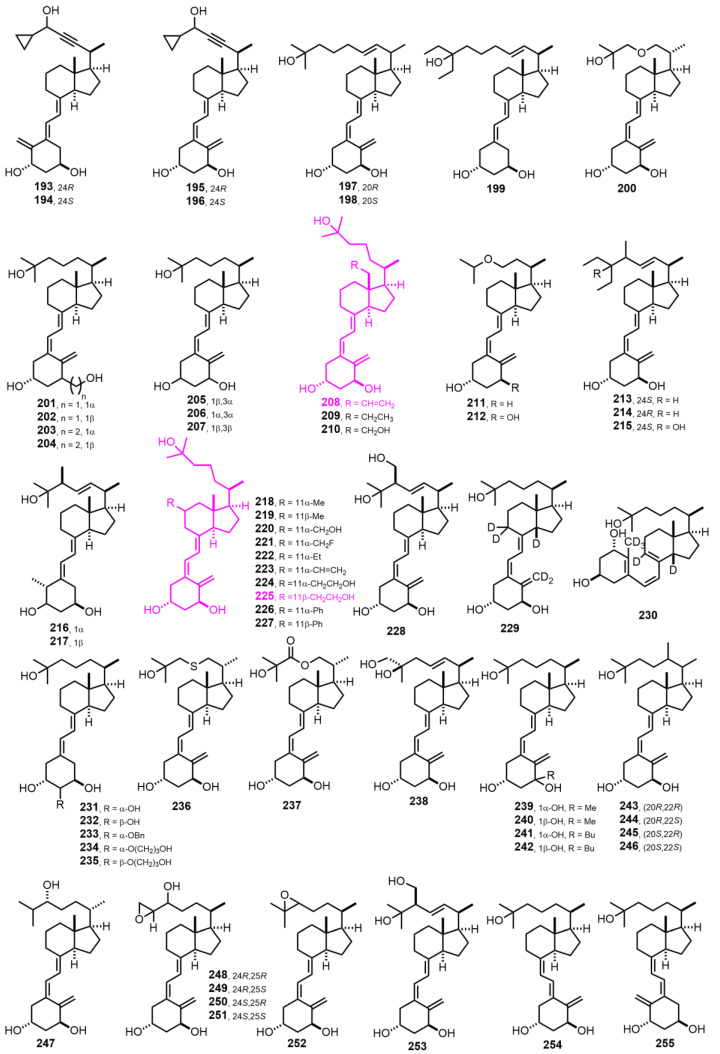
(1993–1994) [118,119,120,121,122,123,124,125,126,127,128,129,130,131,132,133,134,135,136].

**Figure 7 nutrients-14-04927-f007:**
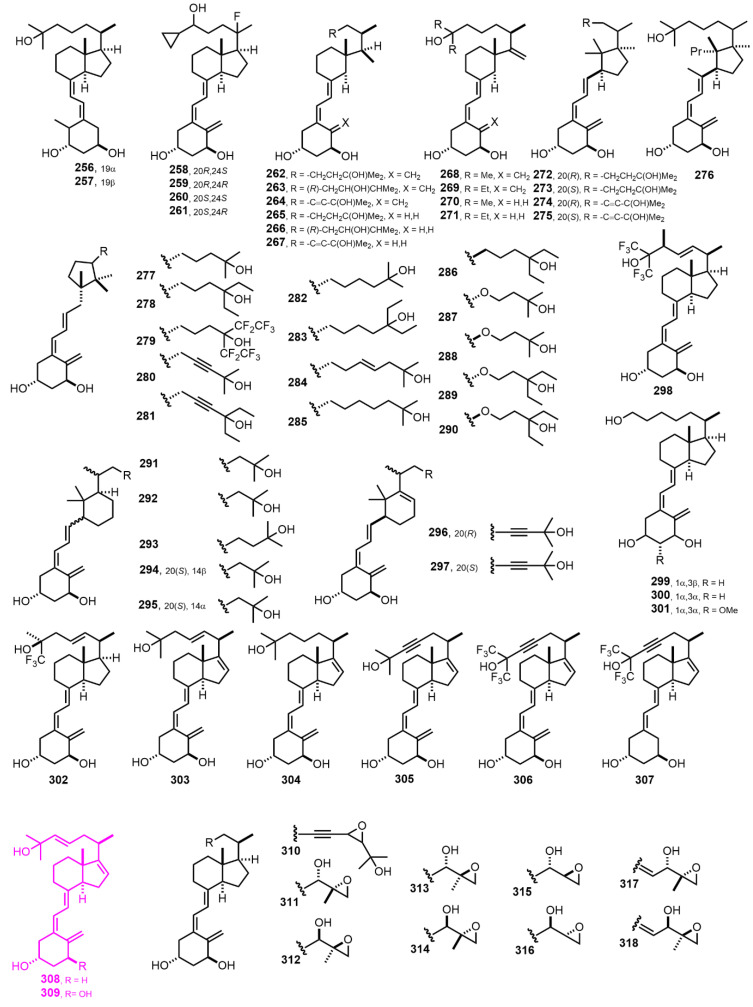
(1994–1997) [136,137,138,139,140,141,142,143,144,145,146,147,148].

**Figure 8 nutrients-14-04927-f008:**
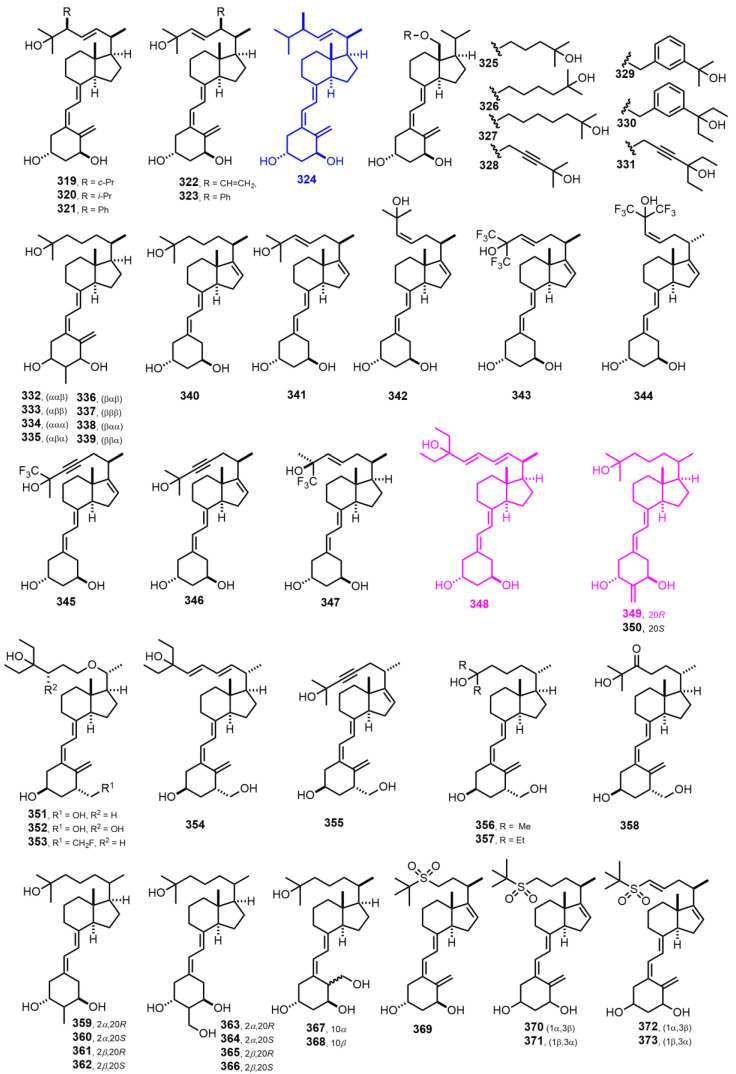
(1997–1999) [149,150,151,152,153,154,155,156,157,158].

**Figure 9 nutrients-14-04927-f009:**
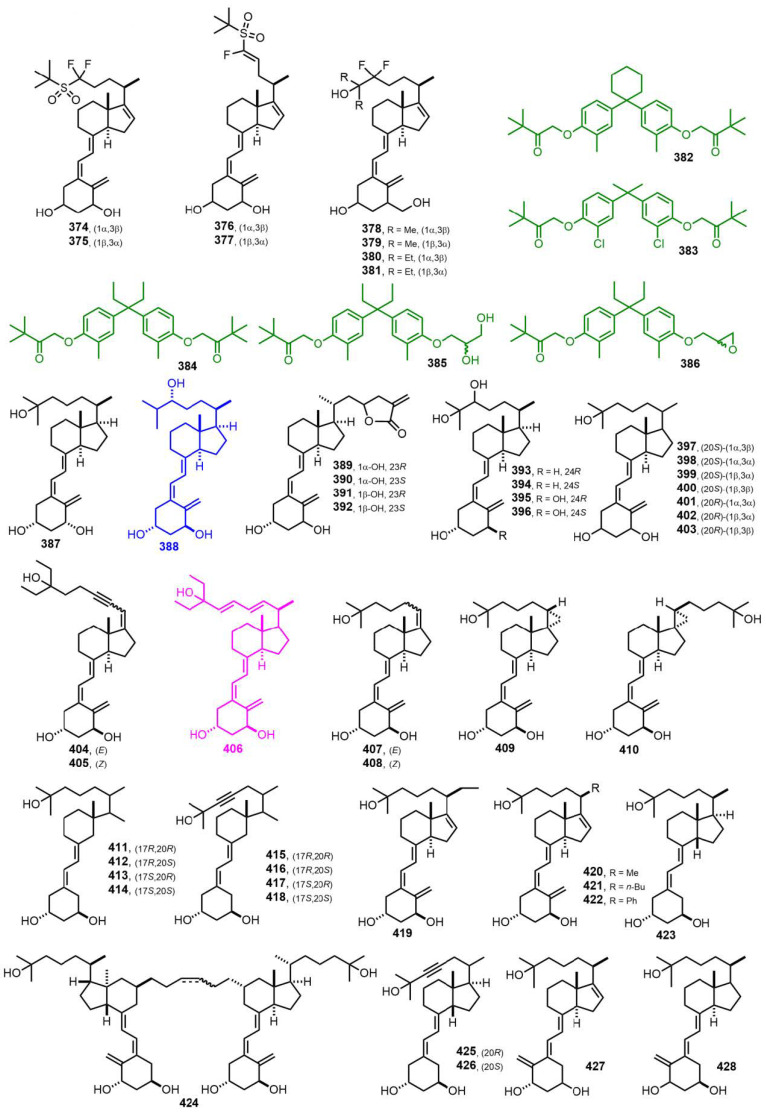
(1999) [158,159,160,161,162,163,164,165,166,167,168].

**Figure 10 nutrients-14-04927-f010:**
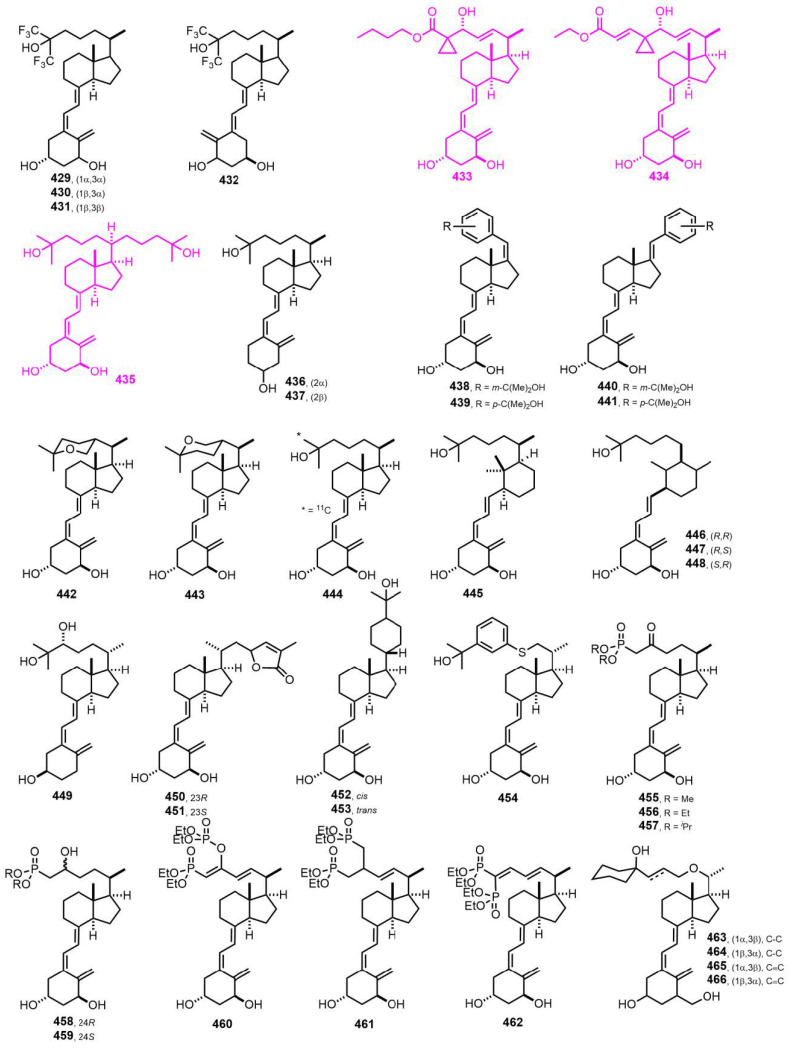
(2000–2001) [169,170,171,172,173,174,175,176,177,178,179,180,181,182].

**Figure 11 nutrients-14-04927-f011:**
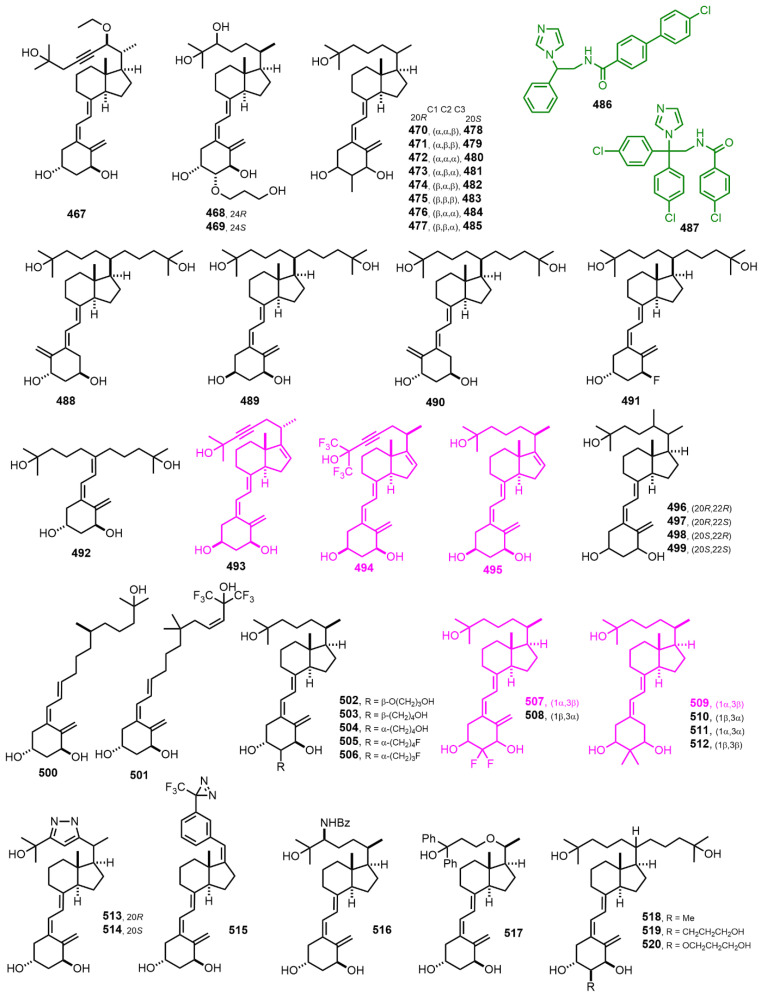
(2001–2002) [183,184,185,186,187,188,189,190,191,192,193,194,195,196].

**Figure 12 nutrients-14-04927-f012:**
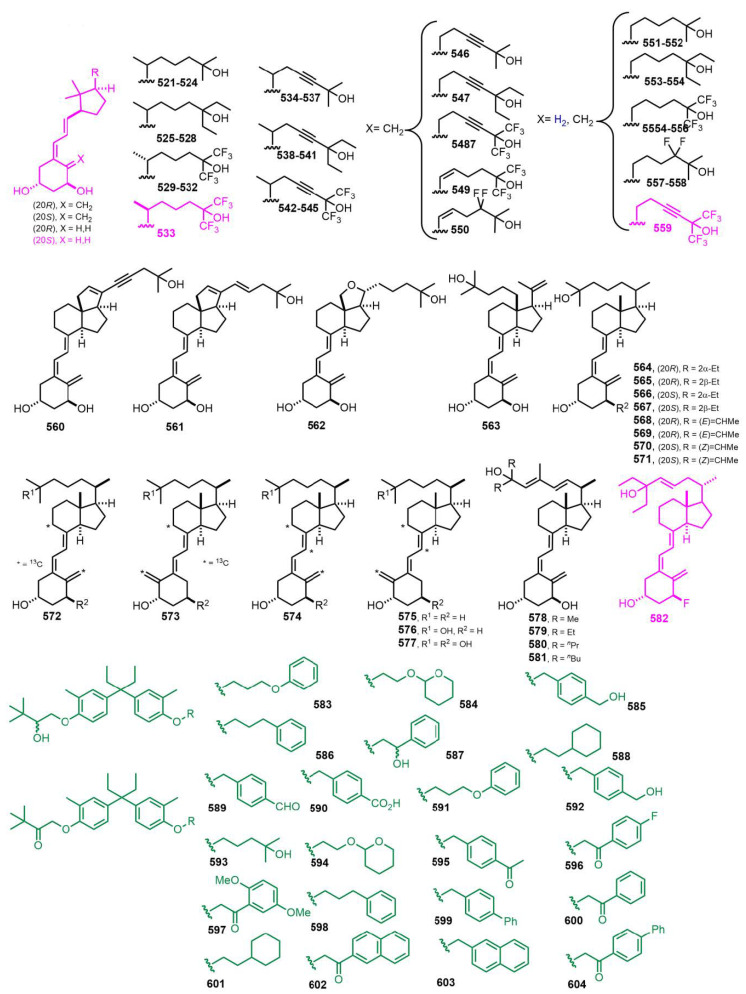
(2002) [197,198,199,200,201,202,203,204].

**Figure 13 nutrients-14-04927-f013:**
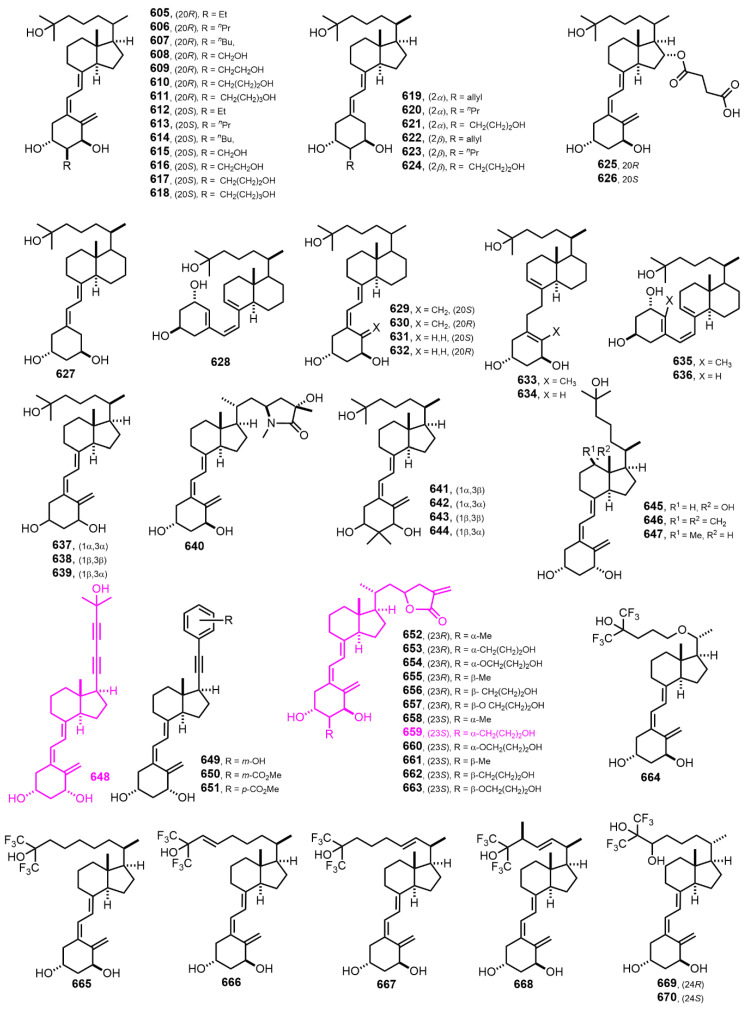
(2003–2004) [205,206,207,208,209,210,211,212,213,214,215,216,217,218].

**Figure 14 nutrients-14-04927-f014:**
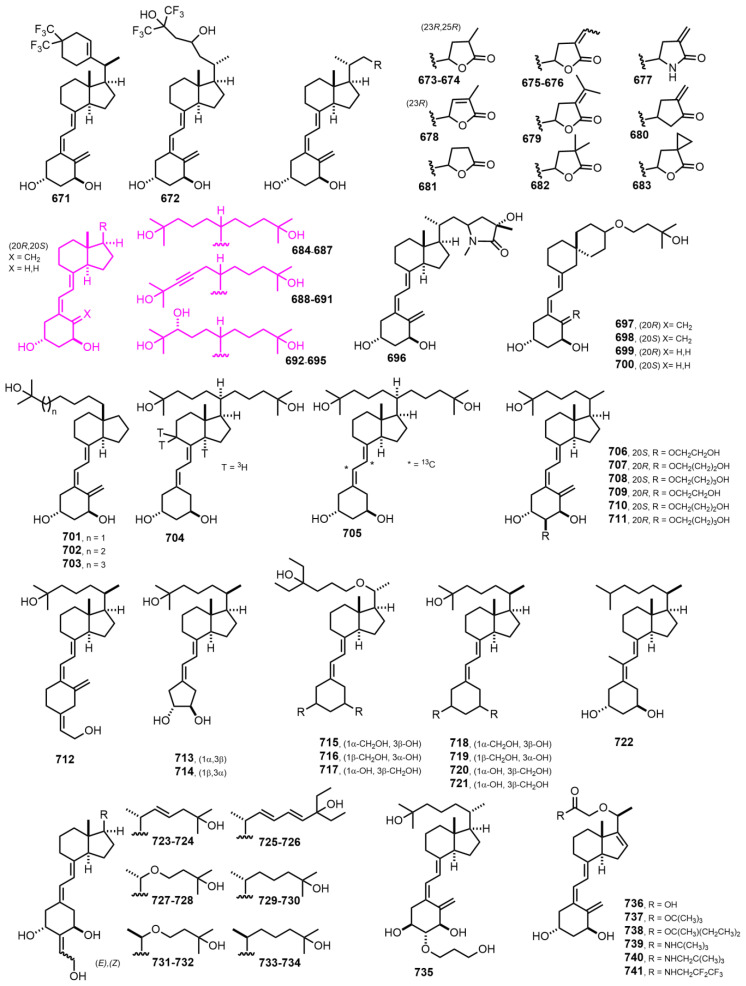
(2004–2006) [218,219,220,221,222,223].

**Figure 15 nutrients-14-04927-f015:**
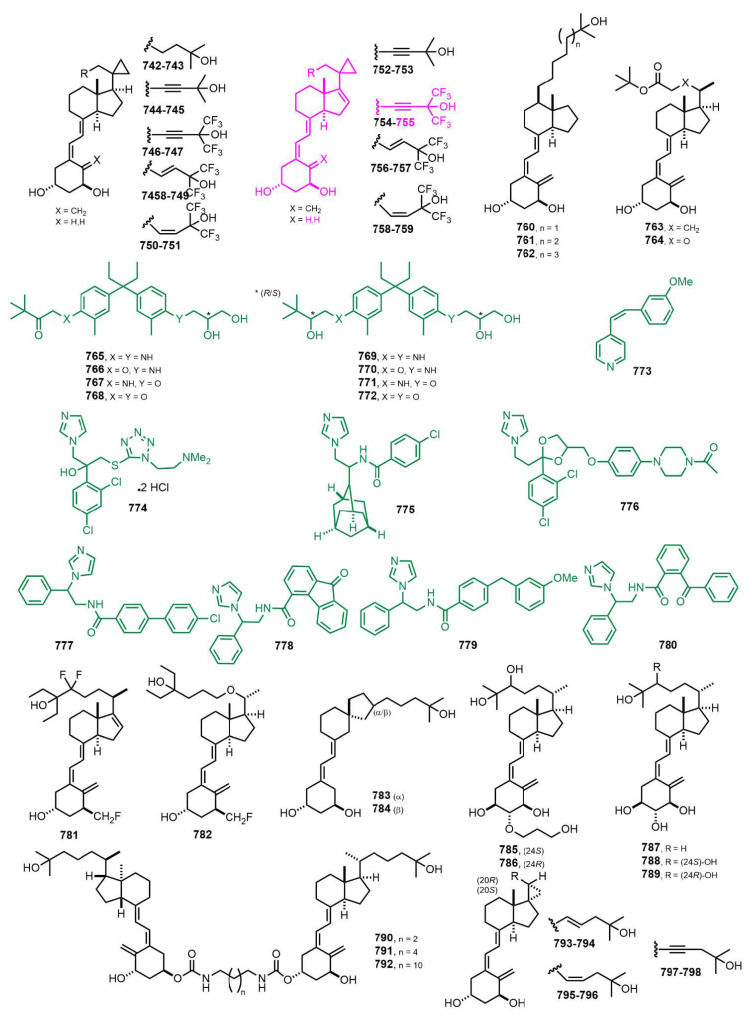
(2006–2007) [224,225,226,227,228,229,230,231,232,233,234,235,236,237,238,239,240].

**Figure 16 nutrients-14-04927-f016:**
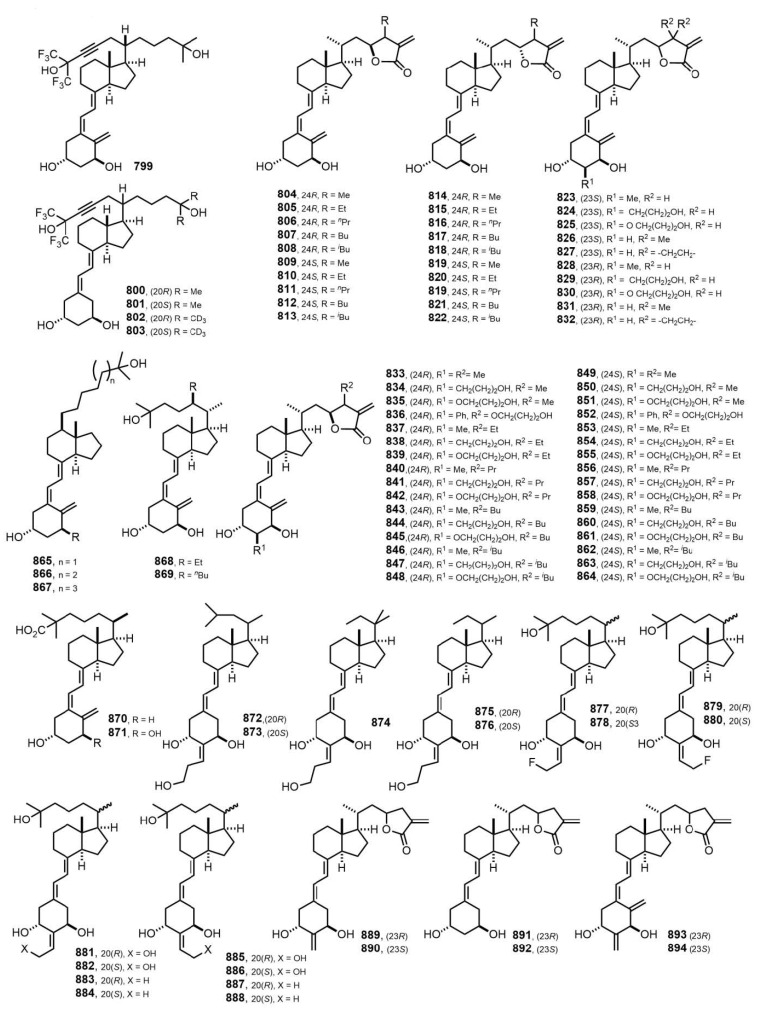
(2006–2008) [241,242,243,244,245,246,247,248,249,250,251,252,253].

**Figure 17 nutrients-14-04927-f017:**
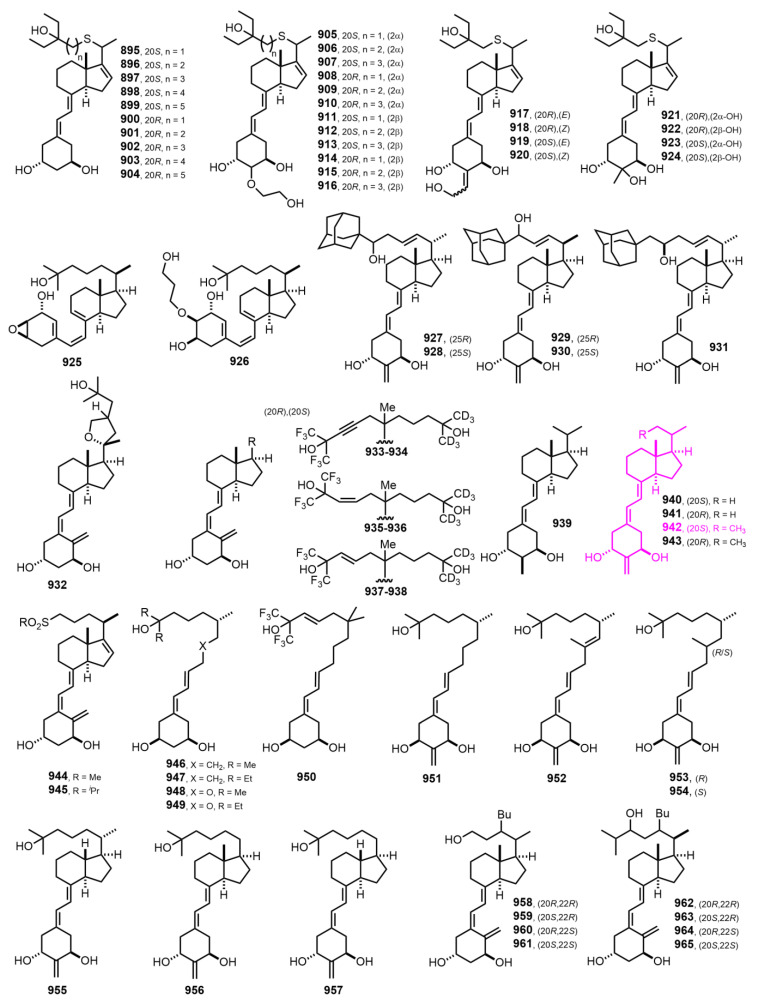
(2008–2009) [254,255,256,257,258,259,260,261,262,263,264].

**Figure 18 nutrients-14-04927-f018:**
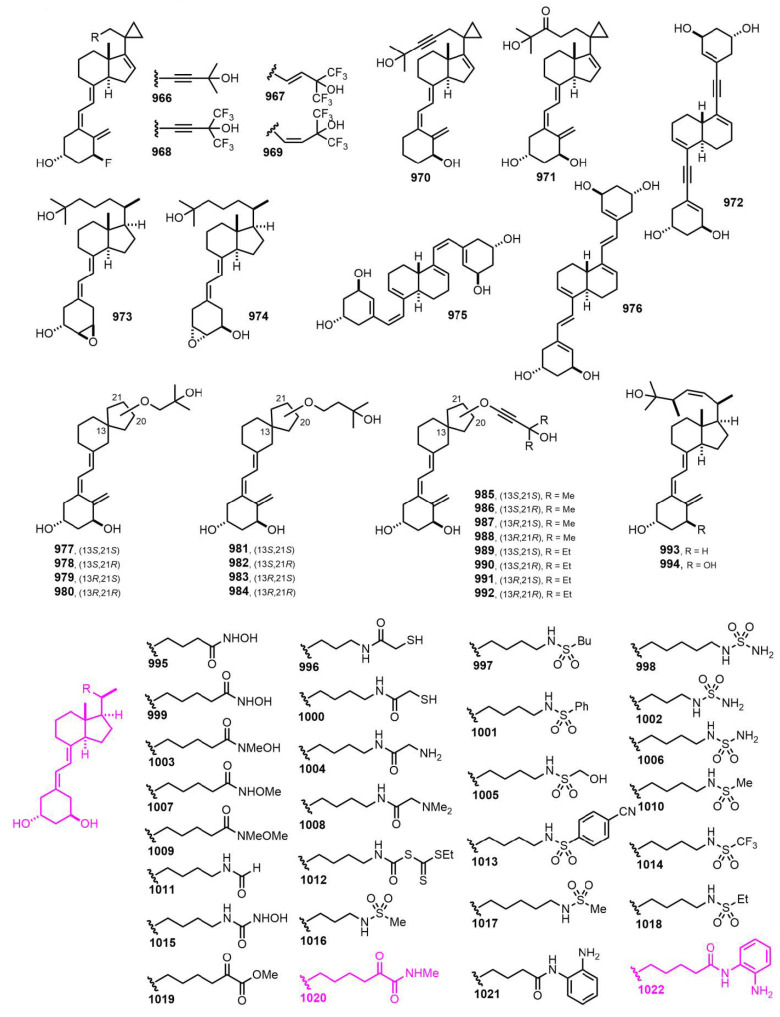
(2009–2010) [265,266,267,268,269,270].

**Figure 19 nutrients-14-04927-f019:**
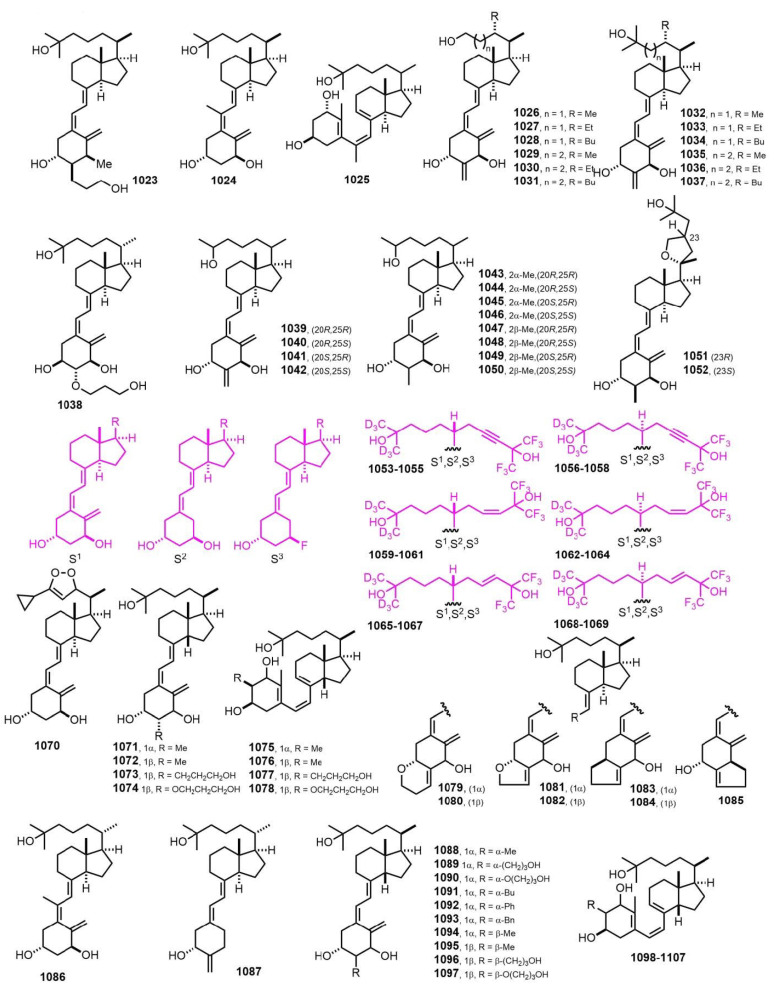
(2009–2010) [271,272,273,274,275,276,277,278,279,280,281,282,283].

**Figure 20 nutrients-14-04927-f020:**
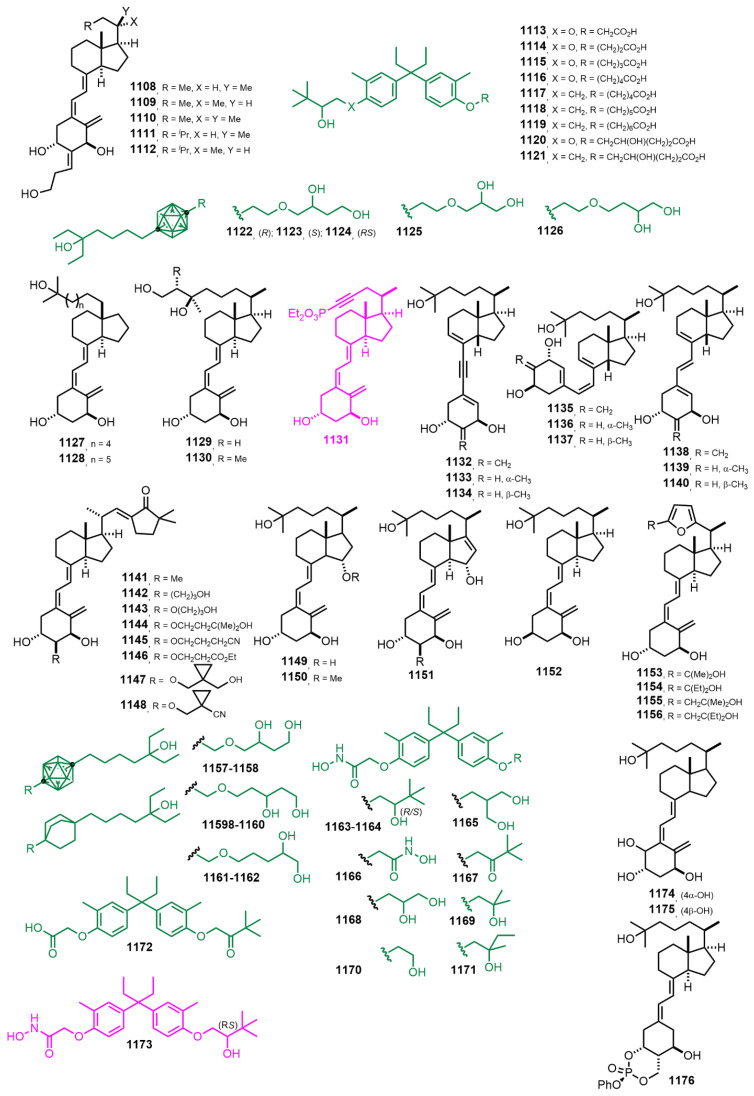
(2010–2012) [284,285,286,287,288,289,290,291,292,293,294,295,296,297,298].

**Figure 21 nutrients-14-04927-f021:**
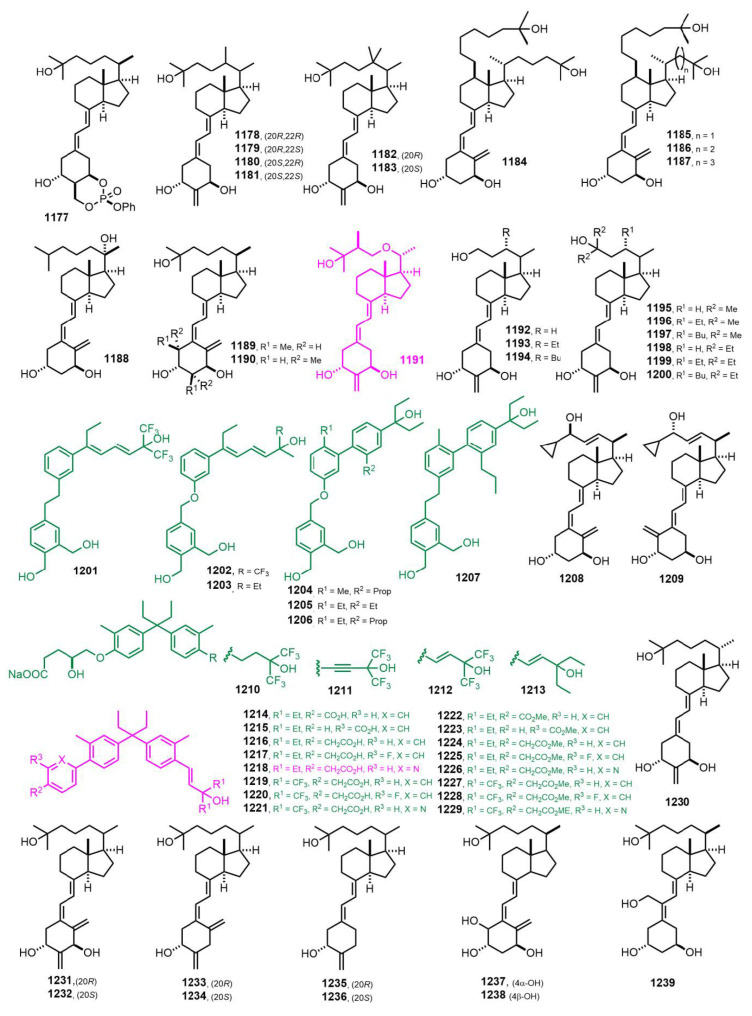
(2012–2013) [298,299,300,301,302,303,304,305,306,307,308,309,310,311,312,313].

**Figure 22 nutrients-14-04927-f022:**
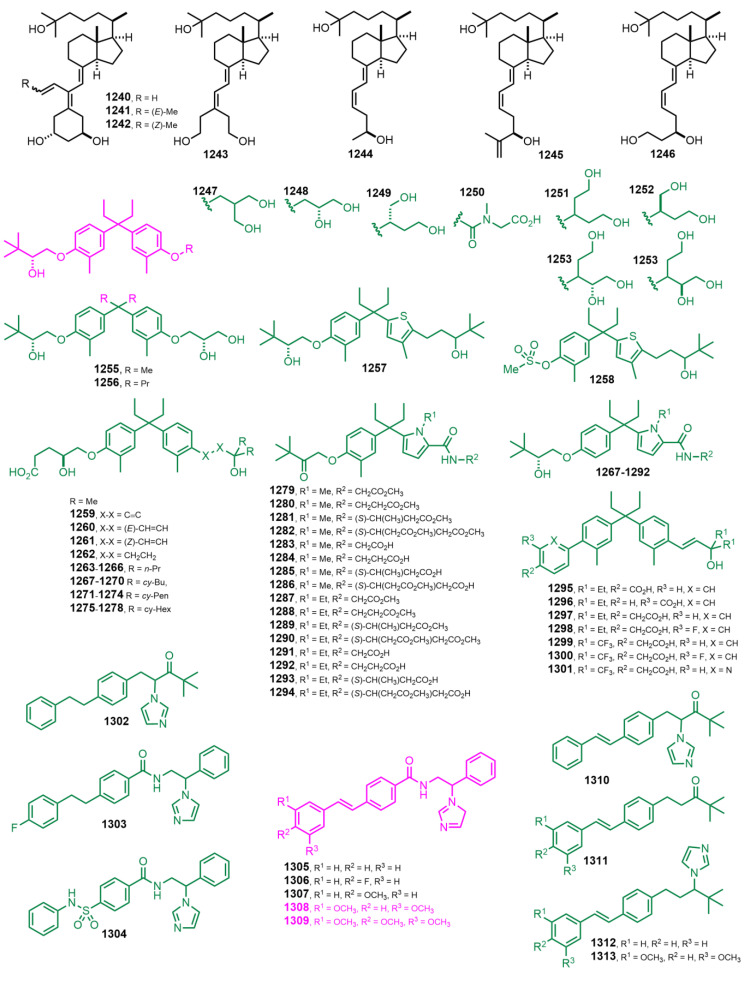
(2013–2014) [313,314,315,316].

**Figure 23 nutrients-14-04927-f023:**
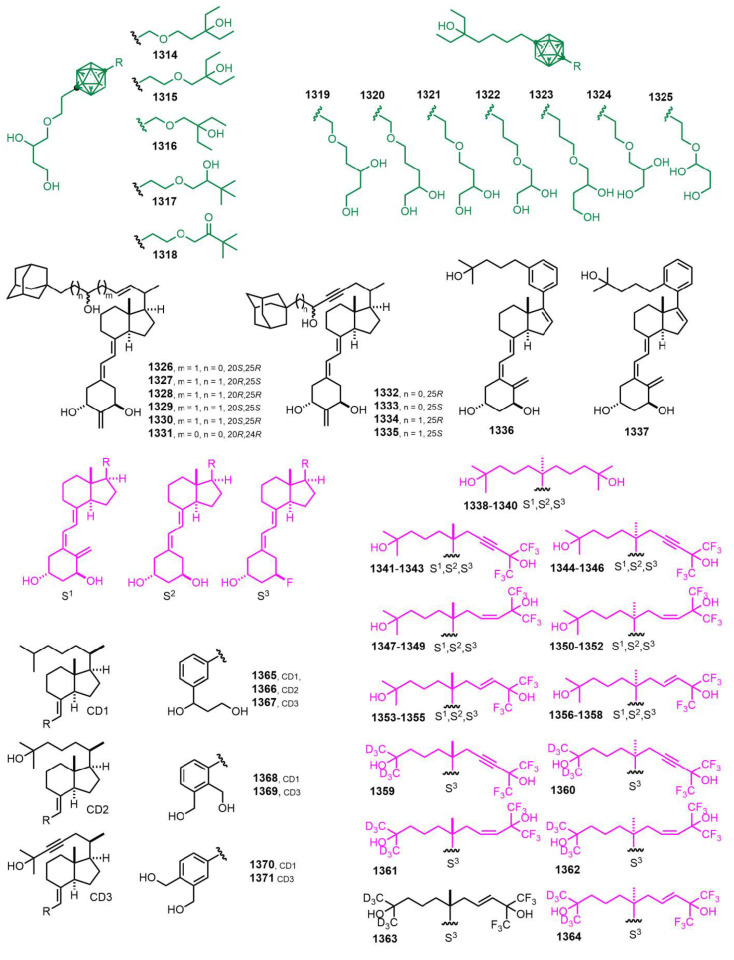
(2014) [317,318,319,320,321].

**Figure 24 nutrients-14-04927-f024:**
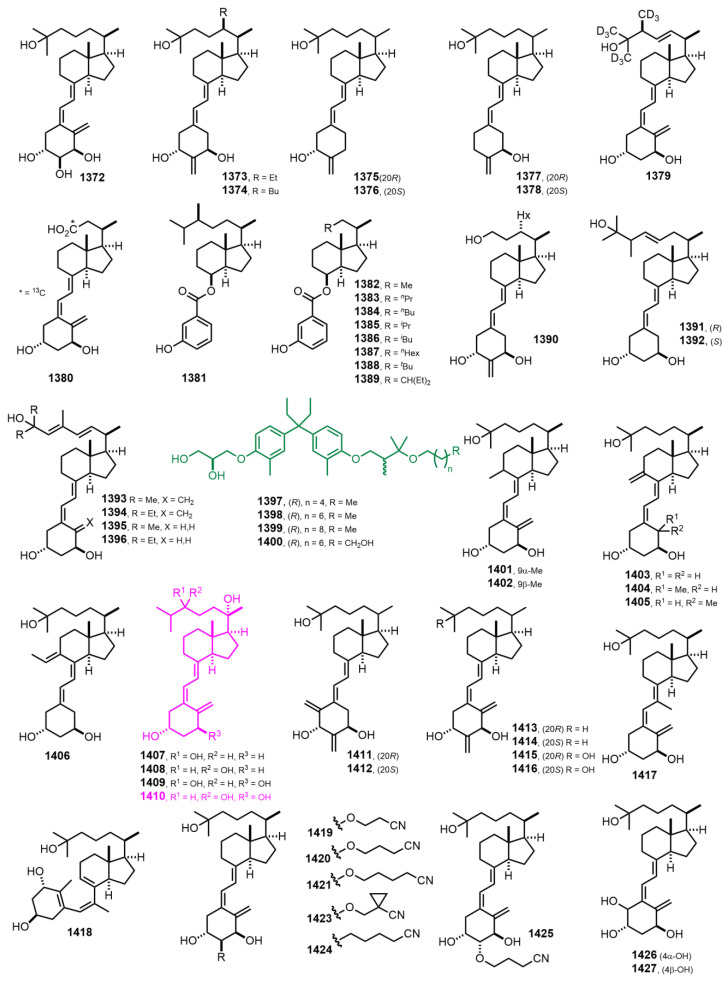
(2014–2015) [322,323,324,325,326,327,328,329,330,331,332,333,334,335,336].

**Figure 25 nutrients-14-04927-f025:**
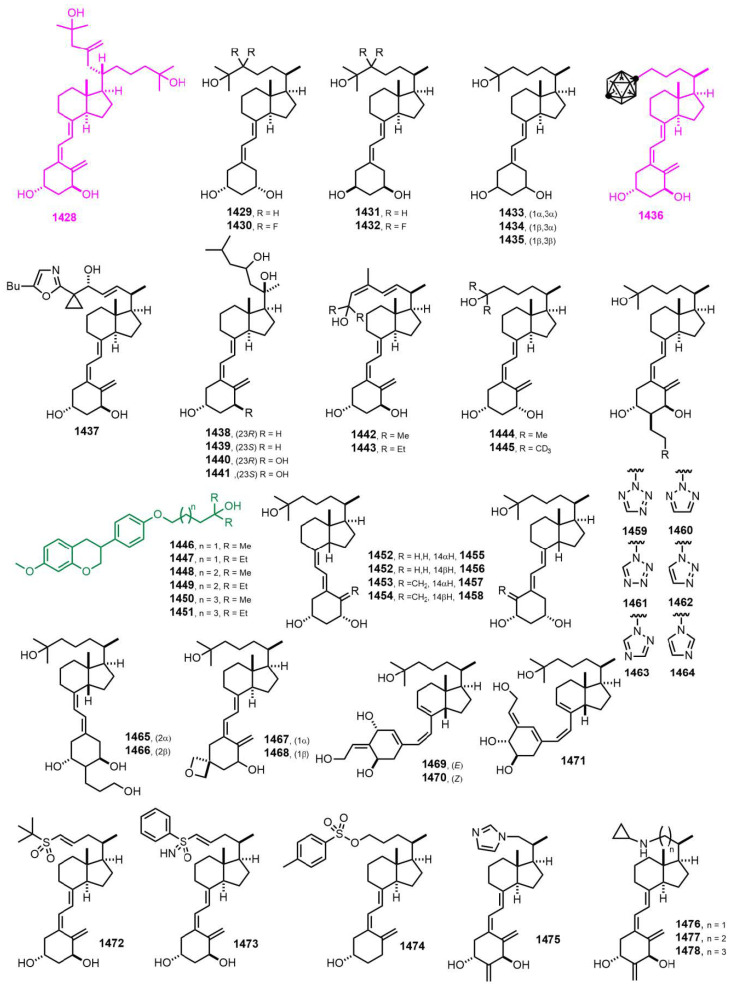
(2015–2017) [337,338,339,340,341,342,343,344,345,346,347,348,349,350,351].

**Figure 26 nutrients-14-04927-f026:**
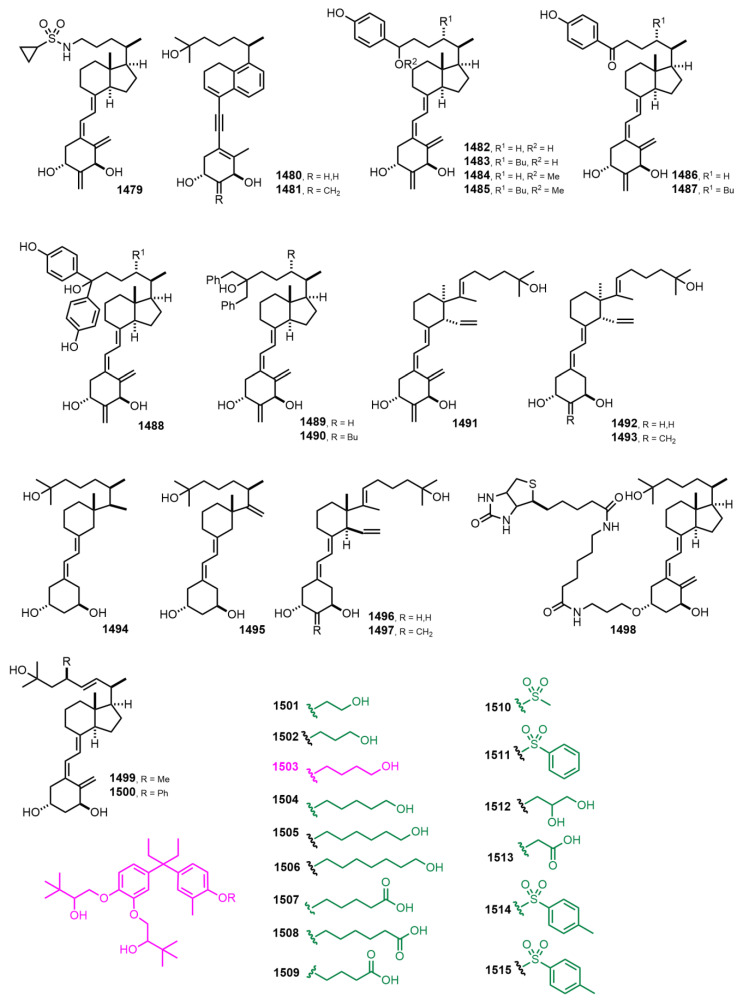
(2017–2018) [351,352,353,354,355,356,357,358].

**Figure 27 nutrients-14-04927-f027:**
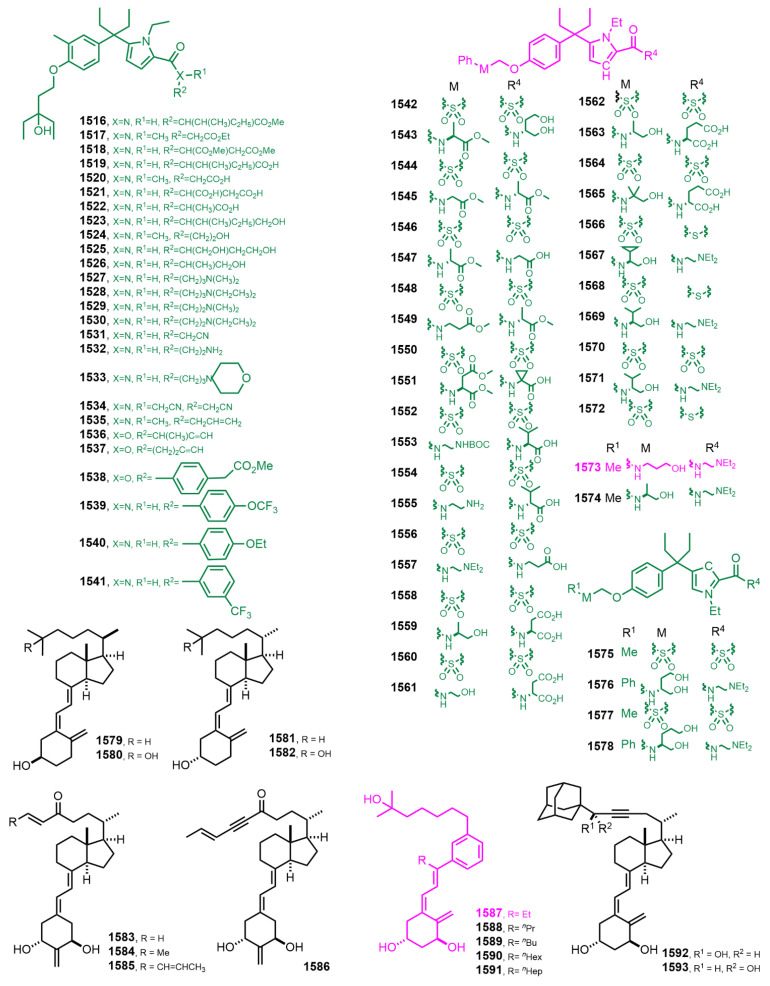
(2018) [359,360,361,362,363,364].

**Figure 28 nutrients-14-04927-f028:**
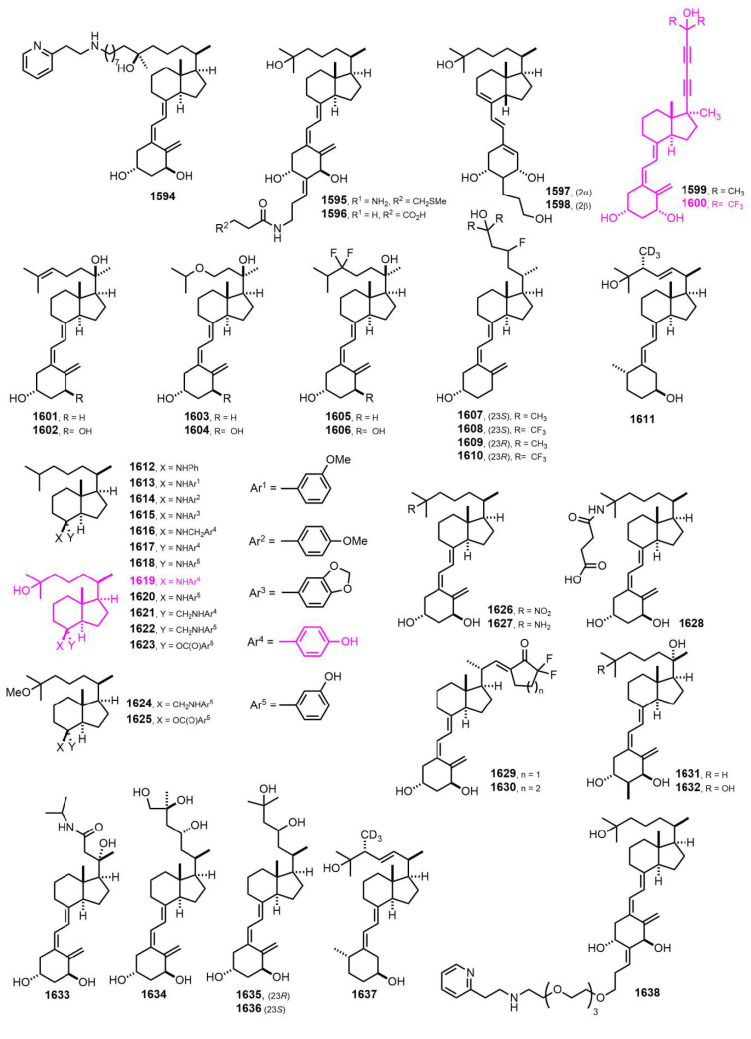
(2018–2019) [365,366,367,368,369,370,371,372,373,374,375,376,377,378].

**Figure 29 nutrients-14-04927-f029:**
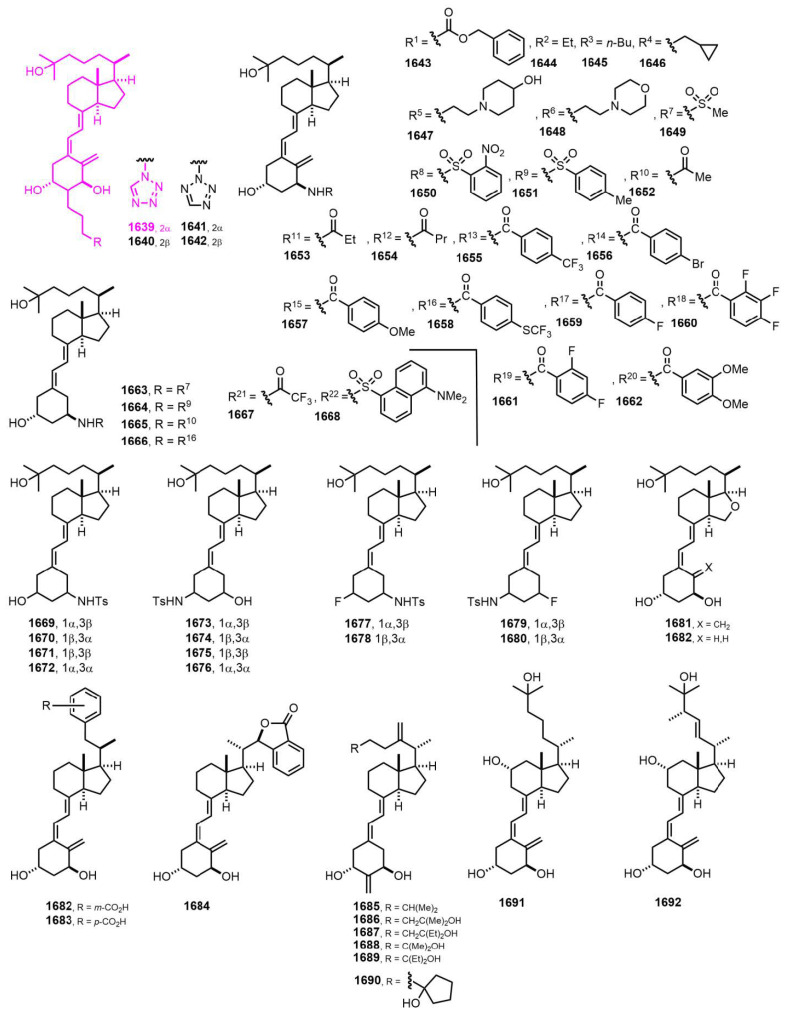
(2019–2020) [379,380,381,382,383].

**Figure 30 nutrients-14-04927-f030:**
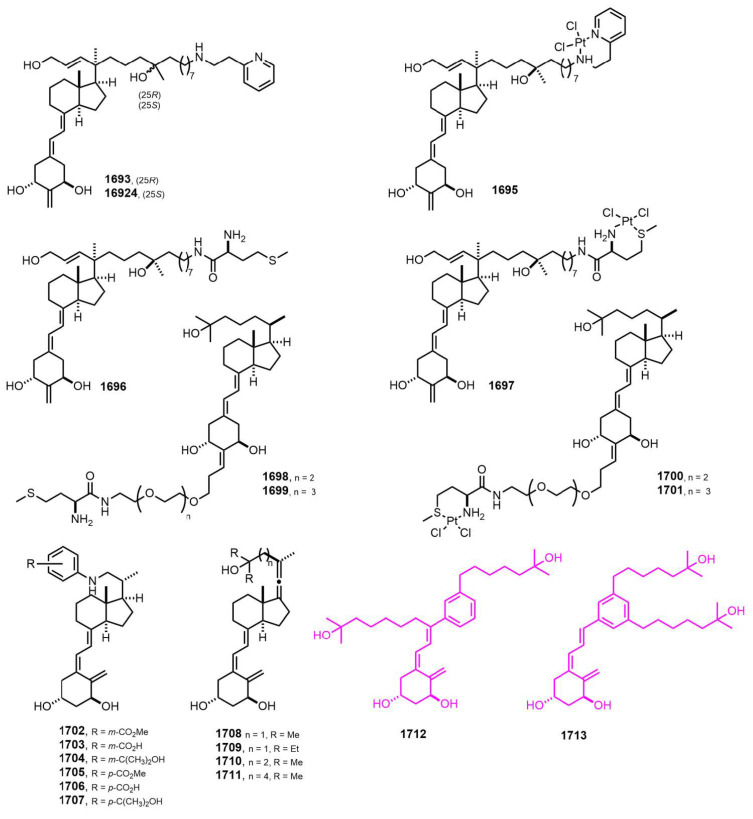
(2020–2022) [384,385,386,387,388,389].

**Figure 31 nutrients-14-04927-f031:**
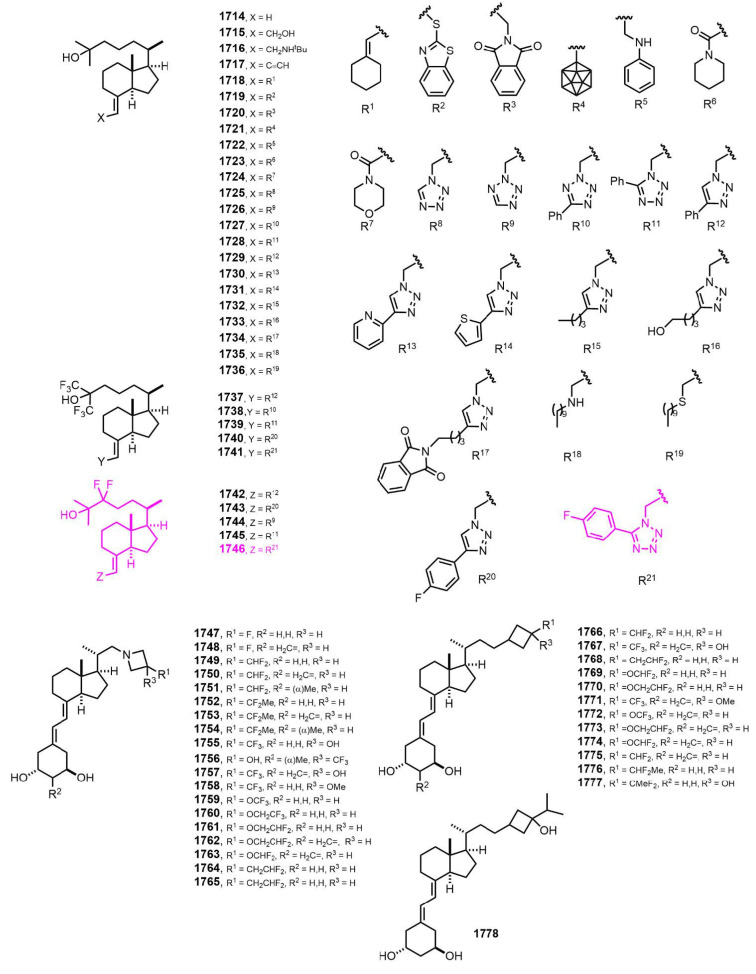
(2021–2022) [390,391,392].

## 4. Conclusions

A century has passed since vitamin D was discovered. The structural diversity achieved among vitamin D receptor ligands (1785 ligands involving metabolites, analogs, hybrids, and nonsteroidal ligands). Seeing as vitamin D plays a ubiquitous role in human physiology, VDR ligands have been found to cure or ameliorate the symptoms of various diseases. It is disheartening to note that for more than twenty years no drug based on a VDR ligand (i.e., analogues, hybrids, or nonsteroidal ligands) has been placed on the market because the structural diversity achieved in the VDR ligands might encode new therapies for other illness different than the calcium–phosphorous homeostasis.

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
