# Peer review of "The Centennial Collection of VDR Ligands: Metabolites, Analogs, Hybrids and Non-Secosteroidal Ligands"

_nutrients, 2022, doi:10.3390/nu14224927_

Round 1
Reviewer 1 Report
This is an interesting review manuscript, but compound numbers and/or reference numbers are incorrect throughout the text, Schemes, and References. Some parts are correct.
Review article requires accuracy in references, especially.
The author should read and check these again carefully before publication.
Author Response
Thank you for your advice.
We have double checked carefully all numbers and references.
Reviewer 2 Report
The text should be corrected by a native English speaker. Moreover also a researcher with biological background should be included to elaborate more on the mechanism of action of vitamin D analogs en to rephrase the parts of text with focus on the biological action of these compounds.
For example:
Line 28 : There is a main draw- back in its therapeutic use, the pharmacological doses employed produce hypercalcemic reactions": not 'produce' but 'induce'
Line 25:' 1,25(OH)2D3 is a pluripotent hormone with strong influence in human health; it is a key controller in musculoskeletal system’.
'it is a key controller in musculoskeletal system' should be replaced by ’1,25(OH)2D3 (1,25D3) plays an essential role in calcium and bone homeostasis.'
Line 25: "Innate immunity system is also positively influenced by normal 1,25(OH)2D3 level in blood."
Innate immune system is "positively" influcenced by …: what does the author mean with “positively”?
Line 26:" Its cell anti-proliferative and pro-differentiation properties have increased the interest of pharma companies.":
explain more in detail antiproliferative and pro-differentiating effects...on which type of cells etc
Line 28 : There is a main draw- back in its therapeutic use, the pharmacological doses employed produce hypercalcemic reactions": not 'produce' but 'induce'
Line 35-35 “Present review updates the scientific information about 35 VDR ligands structural collection, incorporating for first time non-steroidal VDR ligands.”: ‘for the first time’ should be deleted because there exist other papers
Line 48: …through a genomic process: explain?
Line 59-62: ‘The influence over calcemic properties’ …should be replaced by ‘the lower calcemic side effects of vitamin D analogs in comparison with 1,25D3 are important but also other modifications could impact the antiproliferative and pro-differentiating effects or the metabolic stability of 1,25D3.
Line 66: correct 'delection' into 'deletion'
Author Response
I have followed the main siggestions from reviewer #2:
I have included Samuel Seoane, he is a biologist and he has experience in vitamin D physiology. He has corrected all manuscript according to his knowledge.
The whole manuscript has been checked and pulished by a antive English speaker.
L.25. We have changed the sentece "1,25(OH)2D3 is a pluripotent hormone with strong influence in human health; it is a key controller in musculoskeletal system".
L.25. We have changed"positevely"
L.28. We changed "produce hypercalcemic" by "induce hypercalcemic"
L.28 We have rewritten the whole sentece.
L.35. We have deleted "for first time".
L.48 We have rewritten the whole sentece to avoid the expression " through a genomic process".
L.59-62: The paragraph has been rewritten.
L.66. We have corrected the word "delection" by "deletion".
Reviewer 3 Report
The paper by Miguel Maestro suits very well to the collection of papers dedicated to the hundredth anniversary of the discovery of vitamin D. The author collected data about all analogues of vitamin D, and about the ligands of vitamin D receptor that are not related to vitamin D, that have been synthesized throughout these hundred years. This was a valuable job.
Unfortunately, the paper has been written in a way that is not reader friendly, and must be improved.
The schemes have been organized in a chronological way, and this is good. However, descriptions are definitely too short. Each scheme should be followed by a paragraph in which the author should describe properties of the most interesting compounds from given scheme, including his own opinions.
The important issue is the quality of English language. There are many errors, wrong grammar, omissions etc.
For example:
Lines in 53-57 the sentence is hard to understand: There is no scientific rational for the calcemic properties of a compound, structure-function relationships (SAR) were carried out in order to elucidate key modifications in the structure of 1,25(OH)2D3 to alter the biological properties and after more than 50 years, some hints have been obtained.
In line 59 something is missing: calcemic analogs;…
In line 77 the abbreviation is different than elsewhere: l,25(OH),D,
In lines 104-105: instead of “indepently developed for different research groups”, should be “independently developed by different research groups”.
Author Response
We have followed the main suggestions from reviewer #3:
I have included Samuel Seoane, he is a biologist and he has experience in vitamin D physiology. He has corrected all manuscript according to his knowledge.
The whole manuscript has been checked and pulished by a antive English speaker.
Although I understand the reviewer's comment about how Figures have been organized and where the descriptions should be, We have increased the explanations, including our own opinnion but explanations are just ahead Figures because following your comment Figures should be redrawn according to comment extension. And this is a enormous job and it would destroy the actual similar Figures.
Lines 53-57. It has been rewritten.
Line 59: It has been rewritten.
Line 77: 1,25(OH)D3 has been used along the text
Lines 104-105 have been rewitten
Round 2
Reviewer 1 Report
The revised manuscript has still some incorrect relationship between the reference numbers and text description, for example in page 4, Figure 16 covers Gemini and also lactone compounds, but the refs are different from the text citation.
Please check these and whole texts in the manuscript again carefully, figures, texts, and refs. also in the other pages.
Author Response
References and numbering where rechecked through all manuscript.
Reviewer 2 Report
The authors adapted all comments and I fully agree to publish this paper on vitamin D analogs.
Author Response
Thank you
Reviewer 3 Report
The second version of the paper by Maestro and Seoane has improved, but it is still hard to read. In addition, the new text has brought some new problems.
It is good that the paper has an abstract now, but there is a mental shortcut in the abstract that must be corrected: “An enormous effort has been made to synthesize compounds with specific properties while lowering the level of calcium serum.”
In lines 39-40, there is a sentence: “The goal is to reduce calcemic activity while maintaining its biological properties.” Calcemic activity of 1,25(OH)2D3 definitely is a biological property.
In lines 60-61, there are sentences: “Vitamin D is closely associated with calcium and phosphorus homeostasis. No scientific rational has yet been found for the calcemic properties of a compound.” This is not the truth. There is an extensive knowledge how 1,25(OH)2D3 regulates calcium-phosphate homeostasis.
In lines 248-249, the enzymes CYP24A1 and CYP27B1 have been mentioned. Unfortunately, their roles in vitamin D metabolism have not been explained in the paper. The authors should either write more about vitamin D metabolism, or at least refer to some other paper in this issue for such information.
Some new abbreviations were introduced into the paper, which were not explained, for example MLR, HDACi, ZBGs, 1,25D, GADD45α and CDKN1A.
I suppose that the authors should include some more of their opinions, for example they should write why ”for more than twenty years no drug based on a VDR ligand (analogues, hybrids or non-steroidal ligands) has been placed on the market.”
Author Response
R: It is good that the paper has an abstract now, but there is a mental shortcut in the abstract that must be corrected: “An enormous effort has been made to synthesize compounds with specific properties while lowering the level of calcium serum.”
We have rephrased the sentence according to reviwer suggeston: “An enormous effort has been made to synthesize compounds which present beneficial properties while attaining lower calcium serum levels than calcitriol.”
R: In lines 39-40, there is a sentence: “The goal is to reduce calcemic activity while maintaining its biological properties.” Calcemic activity of 1,25(OH)2D3 definitely is a biological property.
We have rephrased the sentence: “Novel structures goal is to reduce their calcemic activity in comparison with calcitriol while exerting their interesting biological properties”.
R: In lines 60-61, there are sentences: “Vitamin D is closely associated with calcium and phosphorus homeostasis. No scientific rational has yet been found for the calcemic properties of a compound.” This is not the truth. There is an extensive knowledge how 1,25(OH)2D3 regulates calcium-phosphate homeostasis.
We agree with reviewer’s opinion “There is an extensive knowledge how 1,25(OH)2D3 regulates calcium-phosphate homeostasis “ but we were not refering to calcemic properties of 1,25(OH)2D3. We have changed the sentence to give full understanding: “No scientific rational has yet been found for the calcemic properties of a compound in comparison with calcitriol”
R: In lines 248-249, the enzymes CYP24A1 and CYP27B1 have been mentioned. Unfortunately, their roles in vitamin D metabolism have not been explained in the paper. The authors should either write more about vitamin D metabolism, or at least refer to some other paper in this issue for such information.
We think that vitamin D metabolism is out of manuscript’s objectives. We have inserted the enzyme information. “Compounds 1300-1311 [300] were analyzed for binding affinity and inhibitory activity against CYP24A1 (24-hydroxylase; this mitochondrial protein initiates the degradation of 1a,25(OH)2D3 by hydroxylation of the side chain), and the imidazole styrylbenzamides 1303-1307 were identified as potent inhibitors of CYP24A1, with similar or greater CYP27B1 (1a-hydroxylase; the protein encoded by this gene it hydroxylates 25OHD3 at the 1a-position, producing 1a,25(OH)2D3) selectivity than standard ketoconazole.
R: Some new abbreviations were introduced into the paper, which were not explained, for example MLR, HDACi, ZBG, 1,25D, GADD45α and CDKN1A.
We have inserted the meaning of the acronims: MLR (Mixed Lymphocite Reaction), HDACi (Histone Deacetylases Inhibitor), ZBG (Zinc Binding Group), 1,25D [1a,25(OH)2D3], GADD45α (Growth arrest and DNA damage 45 gen) and CDKN1A (Cyclin-Dependant Kinase Inhibitor 1A gen).
R: I suppose that the authors should include some more of their opinions, for example they should write why ”for more than twenty years no drug based on a VDR ligand (analogues, hybrids or non-steroidal ligands) has been placed on the market.” We have added an explanation about this sentence. “It is disheartening to note that for more than twenty years no drug based on a VDR ligand (analogues, hybrids or non-steroidal ligands) has been placed on the market because the structural diversity achieved in the VDR ligands might encode new therapies for other illness diferent than the calcium-phosphorous homeostasis.”